# Why Generalization in RL is Difficult: Epistemic POMDPs and Implicit Partial Observability

**Dibya Ghosh**[*,1]          **Jad Rahme**[*,2]          **Aviral Kumar**[1]          **Amy Zhang**[1,3]

**Ryan P. Adams**[2]                    **Sergey Levine**[1]

## Abstract

Generalization is a central challenge for the deployment of reinforcement learning (RL) systems in the real world. In this paper, we show that the sequential structure of the RL problem necessitates new approaches to generalization beyond the well-studied techniques used in supervised learning. While supervised learning methods can generalize effectively without explicitly accounting for epistemic uncertainty, we show that, perhaps surprisingly, this is not the case in RL. We show that generalization to unseen test conditions from a limited number of training conditions induces implicit partial observability, effectively turning even fully-observed MDPs into POMDPs. Informed by this observation, we recast the problem of generalization in RL as solving the induced partially observed Markov decision process, which we call the epistemic POMDP. We demonstrate the failure modes of algorithms that do not appropriately handle this partial observability, and suggest a simple ensemble-based technique for approximately solving the partially observed problem. Empirically, we demonstrate that our simple algorithm derived from the epistemic POMDP achieves significant gains in generalization over current methods on the Procgen benchmark suite.

## 1 Introduction

Generalization is a central challenge in machine learning. However, much of the research on reinforcement learning (RL) has been concerned with the problem of optimization: how to master a specific task through online or logged interaction. Generalization to new test-time contexts has received comparatively less attention, although several works have observed empirically [1, 2, 3, 4] that generalization to new situations poses a significant challenge to RL policies learned from a fixed training set of situations. In standard supervised learning, it is known that in the absence of distribution shift and with appropriate inductive biases, optimizing for performance on the training set (i.e., empirical risk minimization) translates into good generalization performance. It is tempting to suppose that the generalization challenges in RL can be solved in the same manner as empirical risk minimization in supervised learning: when provided a training set of contexts, learn the optimal policy within these contexts and then use that policy in new contexts at test-time.

Perhaps surprisingly, we show that such "empirical risk minimization" approaches can be sub-optimal for generalizing to new contexts in RL, even when these new contexts are drawn from the same distribution as the training contexts. As an anecdotal example of why this sub-optimality arises, imagine a robotic zookeeper for feeding otters that must be trained on some set of zoos. When placed

---

*Equal contribution. [1] UC Berkeley, [2] Princeton University, [3] Facebook AI Research. Correspond to: `dibya@berkeley.edu`, `jrahme@math.princeton.edu`.

35th Conference on Neural Information Processing Systems (NeurIPS 2021).

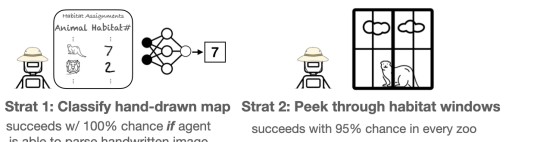 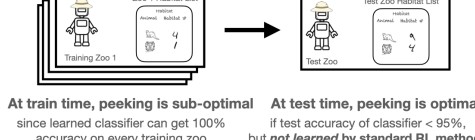

**Strat 1: Classify hand-drawn map**
succeeds w/ 100% chance *if* agent
is able to parse handwritten image

**Strat 2: Peek through habitat windows**
succeeds with 95% chance in every zoo

**At train time, peeking is sub-optimal**
since learned classifier can get 100%
accuracy on every training zoo

**At test time, peeking is optimal**
if test accuracy of classifier < 95%,
but *not learned* by standard RL methods

Figure 1: **Visualization of the robotic zookeeper example.** Standard RL algorithms learn the classifier strategy, since it is optimal in every training zoo, but this strategy is sub-optimal for generalization because peeking generalizes better than the classifier at test-time. This failure occurs due to the following disconnect: while the task is *fully-observed* since the image uniquely specifies the location of the otter habitat, to an agent that has limited training data, the location is *implicitly partially observed at test-time* because of the agent's epistemic uncertainty about the parameters of the image classifier.

in a new zoo, the robot must find and enter the otter enclosure. It can use one of two strategies: either peek through all the habitat windows looking for otters, which succeeds with 95% probability in all zoos, or to follow an image of a hand-drawn map of the zoo that unambiguously identifies the otter enclosure, which will succeed as long as the agent is able to successfully parse the image. In every training zoo, the otters can be found more reliably using the image of the map, and so an agent trained to seek the optimal policy in the training zoos would learn a classifier to predict the identity of the otter enclosure from the map, and enter the predicted enclosure. This classification strategy is optimal on the training environments because the agent can learn to perfectly classify the training zoo maps, but it is *sub-optimal* for generalization, because the learned classifier will never be able to perfectly classify every new zoo map at test-time. Note that this task is *not* partially observed, because the map provides full state information even for a memoryless policy. However, if the learned map classifier succeeds on anything less than 95% of new zoos at test-time, the strategy of peeking through the windows, although always sub-optimal in the training environments, turns out to be a more reliable strategy for finding the otter habitat in a *new* zoo, and results in higher expected returns at test-time.

Although with enough training zoos, the zookeeper can learn a policy by solving the map classification problem, to generalize optimally when given a limited number of zoos requires a more intricate policy that is not learned by standard RL methods. How can we more generally describe the set of behaviors needed for a policy to generalize from a finite number of training contexts in the RL setting? We make the observation that, even in fully-observable domains, the agent's epistemic uncertainty renders the environment *implicitly* partially observed at test-time. In the zookeeper example, although the hand-drawn map provides the exact location of the otter enclosure (and so the enclosure's location is technically fully observed), the agent cannot identify the true parameters of the map classifier from the small set of maps seen at training time, and so the location of the otters is implicitly obfuscated from the agent. We formalize this observation, and show that generalizing optimally at test-time corresponds to solving a partially-observed Markov decision process that we call an **epistemic POMDP**, induced by the agent's epistemic uncertainty about the test environment.

That uncertainty about MDP parameters can be modelled as a POMDP is well-studied in Bayesian RL when training and testing on a single task in an online setting, primarily in the context of exploration [5, 6, 7, 8]. However, as we will discuss, this POMDP interpretation has significant consequences for the generalization problem in RL, where an agent cannot collect more data online, and must instead learn a policy from a fixed set of training contexts that generalizes to new contexts at test-time. We show that standard RL methods that do not explicitly account for this implicit partial observability can be arbitrarily sub-optimal for test-time generalization in theory and in practice. The epistemic POMDP underscores the difficulty of the generalization problem in RL, as compared to supervised learning, and provides an avenue for understanding how we should approach generalization under the sequential nature and non-uniform reward structure of the RL setting. Maximizing expected return in an approximation of the epistemic POMDP emerges as a principled approach to learning policies that generalize well, and we propose LEEP, an algorithm that uses an ensemble of policies to approximately learn the Bayes-optimal policy for maximizing test-time performance.

The primary contribution of this paper is to use Bayesian RL techniques to reframe generalization in RL as the problem of solving a partially observed Markov decision process, which we call the *epistemic POMDP*. The epistemic POMDP highlights the difficulty of generalizing well in RL, as compared to supervised learning. We demonstrate the practical failure modes of standard RL methods, which do not reason about this partial observability, and show that maximizing test-time performance may require algorithms to explicitly consider the agent's epistemic uncertainty during training. Our

work highlights the importance of not only finding ways to help neural networks in RL generalize better, but also on learning policies that degrade gracefully when the underlying neural network eventually does fail to generalize. Empirically, we demonstrate that LEEP, which maximizes return in an approximation to the epistemic POMDP, achieves significant gains in test-time performance over standard RL methods on several ProcGen benchmark tasks.

## 2  Related Work

Many empirical studies have demonstrated the tendency of RL algorithms to overfit significantly to their training environments [1, 2, 3, 4], and the more general increased difficulty of learning policies that generalize in RL as compared to seemingly similar supervised learning problems [9, 10, 11, 12]. These empirical observations have led to a newfound interest in algorithms for generalization in RL, and the development of benchmark RL environments that focus on generalization to new contexts from a limited set of training contexts sharing a similar structure (state and action spaces) but possibly different dynamics and rewards [13, 14, 15, 16, 17].

**Generalization in RL.** Approaches for improving generalization in RL have fallen into two main categories: improving the ability of function approximators to generalize better with inductive biases, and incentivizing behaviors that are easier to generalize to unseen contexts. To improve the representations learned in RL, prior work has considered imitating environment dynamics [18, 19], seeking bisimulation relations [20, 21], and more generally, addressing representational challenges in the RL optimization process [22, 23]. In image-based domains, inductive biases imposed via neural network design have also been proposed to improve robustness to certain factors of variation in the state [24, 25, 26]. The challenges with generalization in RL that we will describe in this paper stem from the deficiencies of MDP objectives, and cannot be fully solved by choice of representations or functional inductive biases. In the latter category, one approach is domain randomization, varying environment parameters such as coefficients of friction or textures, to obtain behaviors that are effective across many candidate parameter settings [27, 28, 29, 30, 31]. Domain randomization sits within a class of methods that seek robust policies by injecting noise into the agent-environment loop, whether in the state [32], the action (e.g., via max-entropy RL) [14], or intermediary layers of a neural network policy (e.g., through information bottlenecks) [22, 33]. In doing so, these methods effectively introduce partial observability into the problem; while not necessarily equivalent to that of the epistemic POMDP, it may indicate why these methods generalize well empirically.

**Bayesian RL:** Our work recasts generalization in RL within the Bayesian RL framework, the problem of acting optimally under a belief distribution over MDPs (see Ghavamzadeh et al. [8] for a survey). Bayesian uncertainty has been studied in many sub-fields of RL [34, 35, 36, 37], the most prominent being for exploration and learning efficiently in the online RL setting. Bayes-optimal behavior in RL is often reduced to acting optimally in a POMDP, or equivalently, a belief-state MDP [6], of which our epistemic POMDP is a specific instantiation. Learning the Bayes-optimal policy exactly is intractable in all but the simplest problems [38, 39], and many works in Bayesian RL have studied relaxations that remain asymptotically optimal for learning, for example with value of perfect information [5, 40] or Thompson sampling [7, 41, 42]. Our main contribution is to revisit these classic ideas in the context of generalization for RL. We find that the POMDP interpretation of Bayesian RL [5, 6, 43] provides new insights on inadequacies of current algorithms used in practice, and explains why generalization in RL can be more challenging than in supervised learning. Being Bayesian in the generalization setting also requires new tools and algorithms beyond those classically studied in Bayesian RL, since test-time generalization is measured using regret over a *single* evaluation episode, instead of throughout an online training process. As a result, algorithms and policies that minimize short-term regret (i.e., are more exploitative) are preferred over traditional algorithms like Thompson sampling that explore thoroughly to ensure asymptotic optimality at the cost of short-term regret.

## 3  Problem Setup

We consider the problem of learning RL policies given a set of training contexts that generalize well to new unseen contexts. This problem can be formalized in a Markov decision process (MDP) where the agent does not have full access to the MDP at training time, but only particular initial states or conditions. Before we describe what this means, we must describe the MDP $\mathcal{M}$, which is given by a tuple $(\mathcal{S}, \mathcal{A}, r, T, \rho, \gamma)$, with state space $\mathcal{S}$, action space $\mathcal{A}$, Markovian transition function $T(s_{t+1}|s_t, a_t)$, bounded reward function $r(s_t, a_t)$, and initial state distribution $\rho(s_0)$. A policy $\pi$ induces a discounted state distribution $d^\pi(s) = (1-\gamma)\mathbb{E}_\pi[\sum_{t\geq 0} \gamma^t 1(s_t = s)]$, and achieves

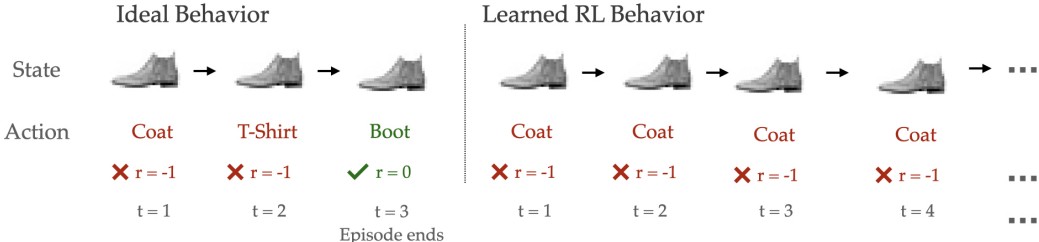

Figure 2: **Sequential Classification RL Problem.** In this task, an agent must keep guessing the label for an image until it gets it correct. To avoid low test return, policies should change actions if the label guessed was incorrect, but standard RL methods fail to do so, instead guessing the same incorrect label repeatedly.

return $J_{\mathcal{M}}(\pi) = \mathbb{E}_{\pi}[\sum_{t\geq 0} \gamma^t r(s_t, a_t)]$ in the MDP. Classical results establish that a deterministic Markovian (memoryless) policy $\pi^*$ maximizes this objective amongst all history-dependent policies.

We focus on generalization in contextual MDPs where the agent is only trained on a training set of contexts, and seeks to generalize well to new contexts. A contextual MDP is an MDP in which the state can be decomposed as $s_t = (c, s_t')$, a context vector $c \in \mathcal{C}$ that remains constant throughout an episode, and a sub-state $s' \in \mathcal{S}'$ that may vary: $\mathcal{S} := \mathcal{C} \times \mathcal{S}'$. Each context vector corresponds to a different situation that the agent might be in, each with slightly different dynamics and rewards, but some shared structure across which an agent can generalize. During training, the agent is allowed to interact only within a sampled subset of contexts $\mathcal{C}_{\text{train}} \subset \mathcal{C}$. The generalization performance of the agent is measured by the return of the agent's policy in the full contextual MDP $J(\pi)$, corresponding to expected performance when placed in potentially new contexts. While our examples and experiments will be in contextual MDPs, our theoretical results also apply to other RL generalization settings where the full MDP cannot be inferred unambiguously from the data available during training, for example in offline reinforcement learning [44].

## 4 Warmup: A Sequential Classification RL Problem

We begin our study of generalization in RL with an example problem that is set up to be close to a supervised learning task where generalization is relatively well understood: image classification on the FashionMNIST dataset [45]. In this environment (visualized in Figure 2), an image from the dataset is sampled (the context) at the beginning of an episode and held fixed; the agent must identify the label of the image to complete the episode. If the agent guesses correctly, it receives a reward of 0 and the episode ends; if incorrect, it receives a reward of $-1$ and the episode continues, so it must attempt another guess for the *same* image at the next time step. This RL problem is near identical to supervised classification, the core distinction being that an agent may interact with the same image over several timesteps in an episode instead of only one attempt as in supervised learning. Note that since episodes may last longer than a single timestep, this problem is not a contextual bandit.

The optimal policy in both the one-step and sequential RL version of the problem deterministically outputs the correct label for the image, because the image fully determines the label (in other words, it is a fully observed MDP). However, this optimal strategy generally cannot be learned from a finite training set, since some generalization error is unavoidable. With a fixed training set, the strategy for generalizing in classification re-

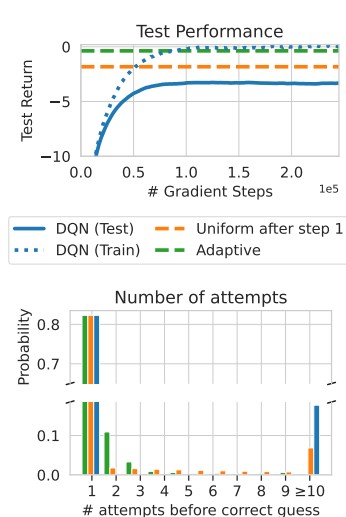

Figure 3: **DQN on RL FashionM-NIST.** DQN achieves lower test performance than simple variants that leverage the structure of the RL problem.

mains the same: deterministically choose the label the agent is most confident about. However, the RL setting introduces two new factors: the agent gets multiple tries at classifying the same image, and it knows if an attempted label is incorrect. To generalize best to new test images, an RL policy must leverage this additional structure, for example by trying many possible labels, or by changing actions if the previous guess was incorrect.

A standard RL algorithm, which estimates the optimal policy on the empirical MDP defined by the dataset of training images does not learn to leverage these factors, and instead learns behavior highly sub-optimal for generalization. We obtained a policy by running DQN [46] (experimental details in Appendix A.1), whose policy deterministically chooses the same label for the image at every timestep. Determinism is not specific to DQN, and is inevitable in any RL method that models the problem as an MDP because the optimal policy in the MDP is always deterministic and Markovian. The learned deterministic policy either guesses the correct label immediately, or guesses incorrectly and proceeds to make the same incorrect guess on every subsequent time-step. We compare performance in Figure 3 with a version of the agent that starts to guess randomly if incorrect on the first timestep, and a different agent that acts by process of elimination: first choosing the action it is most confident about, if incorrect, then the second, and so forth. Although all three versions have the same training performance, the learned RL policy generalizes more poorly than these alternative variants that exploit the sequential nature of the problem. In Section 5.2, we will see that this process-of-elimination is, in some sense, the optimal way to generalize for this task. This experiment reveals a tension: learning policies for generalization that rely on an MDP model fail, even though the underlying environment *is* an MDP. This failure holds in any MDP model with limited data, whether the empirical MDP or more sophisticated MDPs that use uncertainty estimates in their construction.

## 5   Modeling Generalization in RL as an Epistemic POMDP

To better understand test-time generalization in RL, we study the problem under a Bayesian perspective. We show that training on limited training contexts leads to an implicit partial observability at test-time that we describe using a formalism called the epistemic POMDP.

### 5.1   The Epistemic POMDP

In the Bayesian framework, when learning given a limited amount of evidence $\mathcal{D}$ from an MDP $\mathcal{M}$, we can use a prior $\mathcal{P}(\mathcal{M})$ to construct a posterior belief distribution $\mathcal{P}(\mathcal{M}|\mathcal{D})$ over the identity of the MDP. For learning in a contextual MDP, $\mathcal{D}$ corresponds to the environment dynamics and reward in training contexts $\mathcal{C}_{\text{train}}$ that the agent can interact with, and the posterior belief distribution $\mathcal{P}(\mathcal{M}|\mathcal{D})$ models the agent's uncertainty about the behavior of the environment in contexts that it has not seen before (e.g. uncertainty about the label for a test-set image in the example from Section 4).

Since the agent only has partial access to the MDP $\mathcal{M}$ during training, the agent does not know which MDP from the posterior distribution is the true environment, and must act at test-time under this uncertainty. Following a reduction common in Bayesian RL [6, 8], we model this test-time uncertainty using a partially observed MDP that we will call the **epistemic POMDP**. The epistemic POMDP is structured as follows: each new episode in the POMDP begins by sampling a single MDP $\mathcal{M} \sim \mathcal{P}(\mathcal{M}|\mathcal{D})$ from the posterior, and then the agent interacts with $\mathcal{M}$ until the episode ends in this MDP. The agent does not observe *which* MDP was sampled, and since the MDP remains fixed for the duration of the episode, this induces implicit partial observability. Effectively, each episode in the epistemic POMDP corresponds to acting in one of the possible environments that is consistent with the evidence that the agent is allowed access to at training time.

The epistemic POMDP is formally defined as the tuple $\mathcal{M}^{\text{po}} = (\mathcal{S}^{\text{po}}, \mathcal{O}^{\text{po}}, \mathcal{A}, T^{\text{po}}, r^{\text{po}}, \rho^{\text{po}}, \gamma)$. A state in this POMDP $s_t^{\text{po}} = (\mathcal{M}, s_t)$ contains the identity of the current MDP being acted in $\mathcal{M}$, and the current state in this MDP $s_t$; we write the state space as $\mathcal{S}^{\text{po}} = \mathbf{M} \times \mathcal{S}$, where $\mathbf{M}$ is the space of MDPs with support under the prior. The agent only observes $o_t^{\text{po}} = s_t$, the state in the MDP ($\mathcal{O}^{\text{po}} = \mathcal{S}$), but **not** the identity of the MDP, $\mathcal{M}$. The initial state distribution is defined by the posterior distribution: $\rho^{\text{po}}((\mathcal{M}, s_0)) = \mathcal{P}(\mathcal{M}|\mathcal{D})\rho_{\mathcal{M}}(s_0)$, and the transition and reward functions in the POMDP reflect the dynamics in the current MDP:

$$T^{\text{po}}((\mathcal{M}', s') \mid (\mathcal{M}, s), a) = \delta(\mathcal{M}' = \mathcal{M})T_{\mathcal{M}}(s'|s, a) \quad r^{\text{po}}((\mathcal{M}, s), a) = r_{\mathcal{M}}(s, a). \quad (1)$$

**Example** (Sequential Image Classification). We begin by explicitly describing the induced epistemic POMDP for the task from Section 4. The agent's uncertainty concerns how images are mapped to labels, and each MDP $\mathcal{M}$ in the posterior distribution corresponds to a different potential labelling function $Y_{\mathcal{M}} : x \mapsto y$ that is consistent with the training dataset. Each episode in the epistemic POMDP, a different MDP $\mathcal{M}$ and corresponding labeller $Y_{\mathcal{M}}$ is sampled from the posterior distribution, alongside an image $x \sim p(x)$. The agent must guess the label assigned by this labelling function $y := Y_{\mathcal{M}}(x)$, but is only provided the image $x$ and **not** the identity of the labeller $Y_{\mathcal{M}}$. We

emphasize that the context remains *fully observed* in the epistemic POMDP (the image $x$ is provided to the agent); what is partially observed is how the environment dynamics will behave for the context (what label the image corresponds to).

What makes the epistemic POMDP a useful tool for understanding generalization in RL is that performance in the epistemic POMDP $\mathcal{M}^{po}$ corresponds exactly to the expected return of the agent at test-time when the prior is well-specified.

**Proposition 5.1.** *If the true MDP $\mathcal{M}$ is sampled from $\mathcal{P}(\mathcal{M})$, and evidence $\mathcal{D}$ from $\mathcal{M}$ is provided to an algorithm during training, then the expected test-time return of $\pi$ is equal to its performance in the epistemic POMDP $\mathcal{M}^{po}$.*

$$J_{\mathcal{M}^{po}}(\pi) = \mathbb{E}_{\mathcal{M}\sim\mathcal{P}(\mathcal{M})}[J_{\mathcal{M}}(\pi) \mid \mathcal{D}]. \tag{2}$$

*In particular, the optimal policy in $\mathcal{M}^{po}$ is Bayes-optimal for generalization to the unknown MDP $\mathcal{M}$: it receives the highest expected test-time return amongst all possible policies.*

The epistemic POMDP is based on well-understood concepts in Bayesian reinforcement learning, and Bayesian modeling more generally. However, in contrast to prior works on Bayesian RL, we are specifically concerned with settings where there is a training-test split, and performance is measured by a single test episode. While using Bayesian RL to accelerate exploration or minimize regret has been well-explored [8], we rather use the Bayesian lens specifically to understand generalization – a perspective that is distinct from prior work on Bayesian RL. Towards this goal, the equivalence between test-time return and expected return in the epistemic POMDP allows us to use performance in the POMDP as a proxy for understanding how well current RL methods can generalize.

## 5.2   Understanding Optimality in the Epistemic POMDP

We now study the structure of the epistemic POMDP, and use it to characterize properties of Bayes-optimal test-time behavior and the sub-optimality of alternative policy learning approaches. The majority of our results follow from well-known results about POMDPs, so we present them here informally, with formal statements and proofs in Appendix B.

**Example ctd.**   Acting optimally in the epistemic POMDP for the sequential image classification task requires maximizing return over the distribution of labels that is induced by the posterior distribution $p(y|x,\mathcal{D}) = \mathbb{E}_{\mathcal{M}\sim\mathcal{P}(\mathcal{M}|\mathcal{D})}[1(Y_{\mathcal{M}}(x) = y)]$. A deterministic policy (as is learned by standard RL algorithms) is a high-risk strategy in the POMDP; it receives exceedingly low return if the labeller outputs a different label than the one predicted. The Bayes-optimal generalization strategy corresponds to a process of elimination: first choose the most likely label $a = \arg\max p(y|x,\mathcal{D})$; if this is incorrect, eliminate it and choose the next-most likely, repeating until the correct label is finally chosen. Amongst memoryless policies, the optimal behavior is stochastic, sampling actions according to the distribution $\pi^*(a|x) \propto \sqrt{p(y|x,\mathcal{D})}$ (derivation in Appendix A.2).

The characteristics of the optimal policy in the epistemic POMDP for the image classification RL problem match well-established results that optimal POMDP policies are generally memory-based [47], and amongst memoryless policies, the optimal policy may be stochastic [48, 49]. Because of the equivalence between the epistemic POMDP and test-time behavior, these maxims are also true for Bayes-optimal behavior when maximizing test-time performance.

**Remark 5.1.** *The Bayes-optimal policy for maximizing test-time performance is in general non-Markovian. When restricted to Markovian policies, the Bayes-optimal policy is in general stochastic.*

The reason that Bayes-optimal generalization often requires memory is that the experience collected thus far in the episode contains information about the identity of the MDP being acted in (which is hidden from the agent observation), and to maximize expected return, the agent must adapt its subsequent behavior to incorporate this new information. The fact that acting optimally at test-time formally requires adaptivity (or stochasticity for memoryless policies) highlights the difficulty of generalizing well in RL, and provides a new perspective for understanding the success various empirical studies have found in improving generalization performance using recurrent networks [50, 51] and stochastic regularization penalties [32, 14, 22, 33].

It is useful to understand to what degree the partial observability plays a role in determining Bayes-optimal behavior. When the partial observability is insignificant, the epistemic POMDP objective can coincide with a surrogate MDP approximation, and Bayes-optimal solutions can be attained with standard fully-observed RL algorithms. For example, if there is a policy that is simultaneously

optimal in *every* MDP from the posterior, then an agent need not worry about the (hidden) identity of the MDP, and just follow this policy. Perhaps surprisingly, this kind of condition is difficult to relax: we show in Proposition B.1 that even if a policy is optimal in many (but not all) of the MDPs from the posterior, this seemingly "optimal" policy can generalize poorly at test-time.

Moreover, under partial observability, optimal policies for the MDPs in the posterior may differ substantially from Bayes-optimal behavior: in Proposition B.2, we show that the Bayes-optimal policy may take actions that are sub-optimal in *every* environment in the posterior. These results indicate the brittleness of learning policies based on optimizing return in an MDP model when the agent has not yet fully resolved the uncertainty about the true MDP parameters.

**Remark 5.2** (Failure of MDP-Optimal Policies, Propositions B.1, B.2)**.** *The expected test-time return of policies that are learned by maximizing reward in any MDP from the posterior, as standard RL methods do, may be arbitrarily low compared to that of Bayes-optimal behavior.*

As Bayes-optimal memoryless policies are stochastic, one may wonder if simple strategies for inducing stochasticity, such as adding $\epsilon$-greedy noise or entropy regularization, can alleviate the sub-optimality that arose with deterministic policies in the previous paragraph. In some cases, this may be true; one particularly interesting result is that in certain goal-reaching problems, entropy-regularized RL can be interpreted as optimizing an epistemic POMDP objective with a specific form of posterior distribution over reward functions (Proposition B.3) [52]. For the more general setting, we show in Proposition B.4 that entropy regularization and other general-purpose techniques can similarly catastrophically fail in epistemic POMDPs.

**Remark 5.3** (Failure of Generic Stochasticity, Proposition B.4)**.** *The expected test-time return of policies learned with stochastic regularization techniques like maximum-entropy RL that are agnostic of the posterior $\mathcal{P}(\mathcal{M}|\mathcal{D})$ may be arbitrarily low compared to that of Bayes-optimal behavior.*

This failure happens because the degree of stochasticity used by the Bayes-optimal policy reflects the agent's epistemic uncertainty about the environment; since standard regularizations are agnostic to this uncertainty, the learned behaviors often do not reflect the appropriate level of stochasticity needed. A maze-solving agent acting Bayes-optimally, for example, may choose to act deterministically in mazes like those it has seen at training, and on others where it is less confident, rely on random exploration to exit the maze, inimitable behavior by regularization techniques agnostic to this uncertainty.

Our analysis of the epistemic POMDP highlights the difficulty of generalizing well in RL, in the complexity of Bayes-optimal policies (Remark 5.1) and the deficiencies of our standard MDP-based RL algorithms (Remark 5.2, 5.3) . While MDP-based algorithms can serve as a useful starting point for acquiring generalizable skills, learning policies that perform well in new test-time scenarios may require more complex algorithms that attend to the epistemic POMDP structure that is implicitly induced by the agent's epistemic uncertainty.

## 6 Learning Policies that Generalize Well Using the Epistemic POMDP

When the epistemic POMDP $\mathcal{M}^{po}$ can be exactly obtained, we can learn RL policies that generalize well to the true (unknown) MDP $\mathcal{M}$ by learning an optimal policy in the POMDP. In this oracle setting, any POMDP-solving method will suffice, and design choices like policy function classes (e.g. recurrent vs Markovian policies) or agent representations (e.g. belief state vs PSRs) made based on the requirements of the specific domain. However, in practice, the epistemic POMDP can be challenging to approximate due to the difficulties of learning coherent MDP models and maintaining a posterior over such MDP models in high-dimensional domains.

In light of these challenges, we now focus on practical methods for learning generalizable policies when the exact posterior distribution (and therefore true epistemic POMDP) cannot be recovered exactly. We derive an algorithm for learning the optimal policy in the epistemic POMDP induced by an approximate posterior distribution $\hat{\mathcal{P}}(\mathcal{M}|\mathcal{D})$ with finite support. We use this to motivate LEEP, a simple ensemble-based algorithm for learning policies in the contextual MDP setting.

### 6.1 Policy Optimization in an Empirical Epistemic POMDP

Towards a tractable algorithm, we assume that instead of the true posterior $\mathcal{P}(\mathcal{M}|\mathcal{D})$, we only have access to an empirical posterior distribution $\hat{\mathcal{P}}(\mathcal{M}|\mathcal{D})$ defined by $n$ MDP samples from the posterior distribution $\{\mathcal{M}_i\}_{i\in[n]}$. This empirical posterior distribution induces an empirical epistemic POMDP

$\hat{\mathcal{M}}^{\text{po}}$; our ambition is to learn the optimal policy in this POMDP. Rather than directly learning this optimal policy as a generic POMDP solver might, we recognize that $\hat{\mathcal{M}}^{\text{po}}$ corresponds to a collection of $n$ MDPs [2] and decompose the optimization problem to mimic this structure. We will learn $n$ policies $\pi_1, \cdots, \pi_n$, each policy $\pi_i$ in one of the MDPs $\mathcal{M}_i$ from the empirical posterior, and combine these policies together to recover a single policy $\pi$ for the POMDP. Reducing the POMDP policy learning problem into a set of MDP policy learning problems can allow us to leverage the many recent advances in deep RL for scalably solving MDPs. The following theorem links the expected return of a policy $\pi$ in the empirical epistemic POMDP $\hat{\mathcal{M}}^{\text{po}}$, in terms of the performance of the policies $\pi_i$ on their respective MDPs $\mathcal{M}_i$.

**Proposition 6.1.** *Let $\pi, \pi_1, \cdots \pi_n$ be memoryless , and define $r_{\max} = \max_{i,s,a} |r_{\mathcal{M}_i}(s,a)|$. The expected return of $\pi$ in $\hat{\mathcal{M}}^{po}$ is bounded below as:*

$$J_{\hat{\mathcal{M}}^{po}}(\pi) \geq \frac{1}{n} \sum_{i=1}^{n} J_{\mathcal{M}_i}(\pi_i) - \frac{\sqrt{2}r_{\max}}{(1-\gamma)^2 n} \sum_{i=1}^{n} \mathbb{E}_{s \sim d_{\mathcal{M}_i}^{\pi_i}} \left[ \sqrt{D_{KL}\left(\pi_i(\cdot|s) \mid\mid \pi(\cdot|s)\right)} \right], \quad (3)$$

This proposition indicates that if the policies in the collection $\{\pi_i\}_{i \in [n]}$ all achieve high return in their respective MDPs (first term) and are imitable by a single policy $\pi$ (second term), then $\pi$ is guaranteed to achieve high return in the epistemic POMDP. In contrast, if the policies cannot be closely imitated by a single policy, this collection of policies may not be useful for learning in the epistemic POMDP using the lower bound. This means that it may not sufficient to naively optimize each policy $\pi_i$ on its MDP $\mathcal{M}_i$ without any consideration to the other policies or MDPs, since the learned policies are likely to be different and difficult to jointly imitate. To be useful for the lower bound, each policy $\pi_i$ should balance between maximizing performance on its MDP and minimizing its deviation from the other policies in the set. The following proposition shows that if the policies are trained jointly to ensure this balance, it in fact recovers the optimal policy in the empirical epistemic POMDP.

**Proposition 6.2.** *Let $f : \{\pi_i\}_{i \in [n]} \mapsto \pi$ be a function that maps $n$ policies to a single policy satisfying $f(\pi, \cdots, \pi) = \pi$ for every policy $\pi$, and let $\alpha$ be a hyperparameter satisfying $\alpha \geq \frac{\sqrt{2}r_{max}}{(1-\gamma)^2 n}$. Then letting $\pi_1^*, \ldots \pi_n^*$ be the optimal solution to the following optimization problem:*

$$\{\pi_i^*\}_{i \in [n]} = \arg\max_{\pi_1, \cdots, \pi_n} \frac{1}{n} \sum_{i=1}^{n} J_{\mathcal{M}_i}(\pi_i) - \alpha \sum_{i=1}^{n} \mathbb{E}_{s \sim d_{\mathcal{M}_i}^{\pi_i}} \left[ \sqrt{D_{KL}\left(\pi_i(\cdot|s) \mid\mid f(\{\pi_i\})(\cdot|s)\right)} \right], \quad (4)$$

*the policy $\pi^* \coloneqq f(\{\pi_i^*\}_{i \in [n]})$ is optimal for the empirical epistemic POMDP $\hat{\mathcal{M}}^{po}$.*

## 6.2 A Practical Algorithm for Contextual MDPs: LEEP

Proposition 6.2 provides a foundation for a practical algorithm for learning policies when provided training contexts $\mathcal{C}_{\text{train}}$ from an unknown contextual MDP. In order to use the proposition in a practical algorithm, we must discuss two problems: how posterior samples $\mathcal{M}_i \sim \mathcal{P}(\mathcal{M}|\mathcal{D})$ can be approximated, and how the function $f$ that combines policies should be chosen.

**Approximating the posterior distribution:** Rather than directly maintaining a posterior over transition dynamics and reward models, which is especially difficult with image-based observations, we can approximate samples from the posterior via a bootstrap sampling technique [53]. To sample a candidate MDP $\mathcal{M}_i$, we sample with replacement from the training contexts $\mathcal{C}_{\text{train}}$ to get a new set of contexts $\mathcal{C}_{\text{train}}^i$, and define $\mathcal{M}_i$ to be the empirical MDP on this subset of training contexts. Rolling out trials from the posterior sample $\mathcal{M}_i$ then corresponds to selecting a context at random from $\mathcal{C}_{\text{train}}^i$, and then rolling out that context. Crucially, note that $\mathcal{M}_i$ still corresponds to a *distribution* over contexts, not a single context, since our goal is to sample from the posterior entire contextual MDPs.

**Choosing a link function:** The link function $f$ in Proposition 6.2 that combines the set of policies together effectively serves as an inductive bias: since we are optimizing in an approximation to the true epistemic POMDP and policy optimization is not exact in practice, different choices can yield combined policies with different characteristics. Since optimal behavior in the epistemic POMDP must consider all actions, even those that are potentially sub-optimal in all MDPs in the posterior (as discussed in Section 5.2), we use an "optimistic" link function

---

[2]Note that when the true environment is a contextual MDP, the sampled MDP $\mathcal{M}_i$ does not correspond to a single context within a contextual MDP — each MDP $\mathcal{M}_i$ is an *entire* contextual MDP with many contexts.

---

**Algorithm 1** Linked Ensembles for the Epistemic POMDP (LEEP)

---

1: Receive training contexts $\mathcal{C}_{\text{train}}$, number of ensemble members $n$
2: Bootstrap sample training contexts to create $\mathcal{C}_{\text{train}}^1, \dots \mathcal{C}_{\text{train}}^n$, where $\mathcal{C}_{\text{train}}^i \subset \mathcal{C}_{\text{train}}$.
3: Initialize $n$ policies: $\pi_1, \dots \pi_n$
4: **for** iteration $k = 1, 2, 3, \dots$ **do**
5:     **for** policy $i = 1, \dots, n$ **do**
6:         Collect environment samples in training contexts $\mathcal{C}_{\text{train}}^i$ using policy $\pi_i$
7:         Take gradient steps wrt $\pi_i$ on these samples with augmented RL loss:

$$\pi_i \leftarrow \pi_i - \eta \nabla_i (\mathcal{L}^{RL}(\pi_i) + \alpha \mathbb{E}_{s \sim \pi_i, \mathcal{C}_{\text{train}}^i}[D_{KL}(\pi_i(a|s) \| \max_j \pi_j(a|s))])$$

8: Return $\pi = \max_i \pi$: $\pi(a|s) = \frac{\max_i \pi_i(a|s)}{\sum_{a'} \max_i \pi_i(a'|s)}$.

---

that does not dismiss any action that is considered by at least one of the policies, specifically $f(\{\pi_i\}_{i \in [n]}) = (\max_i \pi_i)(a|s) := \frac{\max \pi_i(a|s)}{\sum_{a'} \max \pi_i(a'|s)}$.

**Algorithm:** We learn a set of $n$ policies $\{\pi_i\}_{i \in [n]}$, using a policy gradient algorithm to implement the update step. To update the parameters for $\pi_i$, we take gradient steps via the surrogate loss used for the policy gradient, augmented by a disagreement penalty between the policy and the combined policy $f(\{\pi_i\}_{i \in [n]})$ with a penalty parameter $\alpha > 0$, as in Equation 5:

$$\mathcal{L}(\pi_i) = \mathcal{L}^{RL}(\pi_i) + \alpha \mathbb{E}_{s \sim \pi_i, \mathcal{M}_i}[D_{KL}(\pi_i(a|s) \| \max_j \pi_j(a|s))]. \tag{5}$$

Combining these elements together leads to our method, LEEP, which we summarize in Algorithm 1. In our implementation, we use PPO for $\mathcal{L}^{RL}(\pi_i)$ [54]. In summary, LEEP bootstrap samples the training contexts to create overlapping sets of training contexts $\mathcal{C}_{\text{train}}^1, \dots \mathcal{C}_{\text{train}}^n$. Every iteration, each policy $\pi_i$ generates rollouts in training contexts chosen uniformly from its corresponding $\mathcal{C}_{\text{train}}^i$, and is then updated according to Equation 5, which both maximizes the expected reward and minimizes the disagreement penalty between each $\pi_i$ and the combined policy $\pi = \max_j \pi_j$.

While this algorithm is structurally similar to algorithms for multi-task learning that train a separate policy for each context or group of contexts with a disagreement penalty [55, 56], the motivation and the interpretation of these approaches are completely distinct. In multi-task learning, the goal is to solve a given set of tasks, and these methods promote transfer via a loss that encourages the solutions to the tasks to be in agreement. In our setting, while we also receive a set of tasks (contexts), the goal is not to maximize performance on the training tasks, but rather to learn a policy that maximizes performance on unseen test tasks. The method also has a subtle but important distinction: each of our policies $\pi_i$ acts on a sample from the contextual MDP posterior (which captures epistemic uncertainty), *not* a single training context [55] or element from a disjoint partitioning [56] (which does not). This distinction is crucial, since our generalization performance requires our aim is not to make it easier to solve the training contexts, but the opposite: prevent the algorithm from overfitting to the individual training contexts. Correspondingly, our experiments confirm that such multi-task learning approaches do not provide the same generalization benefits as our approach.

## 7  Experiments

The primary ambition of our empirical study is to test the hypothesis that policies that are learned through (approximations of) the epistemic POMDP do in fact attain better test-time performance than those learned by standard RL algorithms. We do so on the Procgen benchmark [16], a challenging suite of diverse tasks with image-based observations testing generalization to unseen contexts.

1. Does LEEP derived from the epistemic POMDP lead to improved test-time performance over standard RL methods?
2. Can LEEP prevent overfitting when provided a limited number of training contexts?
3. How do different algorithmic components of LEEP affect test-time performance ?

The Procgen benchmark is a set of procedurally generated games, each with different generalization challenges. In each game, during training, the algorithm can interact with 200 training levels, before it is asked to generalize to the full distribution of levels. The agent receives a $64 \times 64 \times 3$ image observation, and must output one of 15 possible actions. We instantiate our method using an ensemble

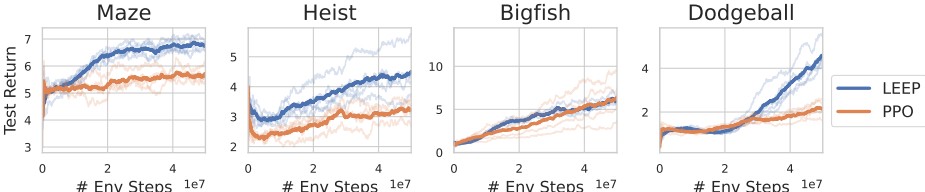

Figure 4: Test set return for LEEP and PPO throughout training in four Procgen environments (averaged across 5 random seeds). LEEP achieves higher test returns than PPO on three tasks (Maze, Heist and Dodgeball) and matches test return on Bigfish while having less variance across seeds.

of $n = 4$ policies, a penalty parameter of $\alpha = 1$, and PPO [54] to train the individual policies (full implementation details in Appendix C).

We evaluate our method on four games in which prior work has found a large gap between training and test performance, and which we therefore conclude pose a significant generalization challenge [16, 23, 26]: Maze, Heist, BigFish, and Dodgeball. In Figure 4, we compare the test-time performance of the policies learned using our method to those learned by a PPO agent with entropy regularization. In three of these environments (Maze, Heist, and Dodgeball), our method outperforms PPO by a significant margin, and in all cases, we find that the generalization gap between training and test performance is lower for our method than PPO (Appendix D.1).

To understand how LEEP behaves with fewer training contexts, we ran on the Maze task with only 50 levels (Figure 5 (top)); the test return of the PPO policy decreases through training, leading to final performance worse than the starting random policy, but our method avoids this degradation.

We perform an ablation study on the Maze and Heist environments (Maze in Figure 4, Heist in Appendix D.1) to rule out potential confounding causes for the improved generalization that our method displays on the Procgen benchmark tasks. First, to see if the performance benefit derives solely from the use of ensembles, we compare LEEP to a Bayesian model averaging strategy that trains an ensemble of policies without regularization ("Ensemble (no reg)"), and uses a mixture of these policies. This strategy does improve performance over the PPO policy, but does not match LEEP, indicating the usefulness of the regularization. Second, we compared to a version of LEEP that combines the ensemble policies together using the average $\frac{1}{n}\sum_{i=1}^{n}\pi_i(a|s)$ ("LEEP (avg)"). This link function achieves worse test-time performance than the optimistic version, which indicates that the inductive bias conferred by the $\max_i \pi_i$ link function is a useful component of the algorithm. We also compare to Distral, a multi-task learning method with different motivations

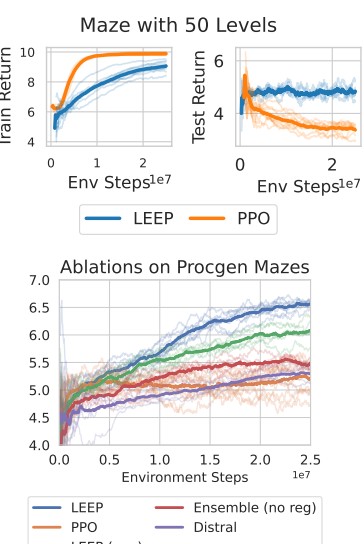

Figure 5: **(top)** Performance of LEEP and PPO with only 50 training levels on Maze. **(bottom)** Ablations of LEEP in Maze.

but similar structure to LEEP: this method helps accelerate learning on the provided training contexts (figures in Appendix D.1), but does not improve generalization performance as LEEP does. We additionally ablated the two key hyperparameters in LEEP, the number of ensemble members $n$ and the penalty coefficient $\alpha$ (Table in Appendix D.2).

## 8 Discussion

It has often been observed experimentally that generalization in RL poses a significant challenge, but it has so far remained an open question as to whether the RL setting itself presents additional generalization challenges beyond those seen in supervised learning. In this paper, we answer this question in the affirmative, and show that, in contrast to supervised learning, generalization in RL results in a new type of problem that cannot be solved with standard MDP solution methods, due to partial observability induced by epistemic uncertainty. We call the resulting partially observed setting the epistemic POMDP, where uncertainty about the true underlying MDP results in a challenging partially observed problem. We present a practical approximate method that optimizes a bound for performance in an approximation of the epistemic POMDP, and show empirically that this approach, which we call LEEP, attains significant improvements in generalization over other RL methods that

do not properly incorporate the agent's epistemic uncertainty into policy optimization. A limitation of this approach is that it optimizes a crude approximation to the epistemic POMDP with a small number of posterior samples, and may be challenging to scale to better approximations to the true objective. Developing algorithms that better model the epistemic POMDP and optimize policies within is an exciting avenue for future work, and we hope that this direction will lead to further improvements in generalization in RL.

## Acknowledgements

This research was supported by an NSF graduate fellowship, the DARPA assured autonomy program, the NSF IIS-2007278 grant, a Princeton SEAS Innovation Grant and compute support from Google and Microsoft. We thank Benjamin Eysenbach, Xinyang Geng, and Justin Fu as well as members of the Princeton Laboratory for Intelligent Probabilistic Systems for helpful discussions and feedback.

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
