# A FashionMNIST Classification

## A.1 Implementation details

**Environment:** The RL image classification environment consists of a dataset of labelled images. At the beginning of each episode, a new image and its corresponding label are chosen from the dataset, and held fixed for the entire episode. Each time-step, the agent must pick an action corresponding to one of the labels. If the picked label is correct, the agent gets a reward of $r = 0$, and the episode ends, and if the picked label is incorrect, then the agent gets a reward of $r = -1$, and the episode continues to the next time-step (where it must guess another label for the *same* image). The total return for a trajectory corresponds to the number of incorrect guesses the agent makes for the image. We enforce a time-limit of 20 timesteps in the environment to prevent infinite-length trajectories of incorrect guessing.

We train the agent on a dataset of 10000 FashionMNIST images subsampled from the training set, and test on the FashionMNIST test dataset. Note that this task is very similar, but not exactly equivalent to maximizing predictive accuracy for supervised classification: if the episode ended regardless of whether or not the agent was correct, then it would correspond exactly to classification.

**Algorithm:** We train a DQN agent on the training environment using the min-Q update rule from TD3 [57]. The Q-function architecture is a convolutional neural network (CNN) with the architecture from Kostrikov et al [58]. To ensure that the agent does not suffer from poor exploration during training, the replay buffer is pre-populated with one copy of every possible transition in the training environment (that is, where every action is taken for every image in the training dataset). The variant labelled "Uniform after step 1" in Figure 3 follows the DQN policy for the first time-step, and if this was incorrect, then at all subsequent time-steps, takes a random action uniformly amongst the 10 labels. For the variant labelled "Adaptive", we train a classifier $p_\theta(y|x)$ on the training dataset of images with the same architecture as the DQN agent. The adaptive agent follows a process-of-elimination strategy; formally, the action taken by the adaptive agent at time-step $t$ is given by $\arg\max_{a \notin \{a_1, \ldots, a_{t-1}\}} p_\theta(y = a|x)$.

## A.2 Derivation of Bayes-optimal policies

In the epistemic POMDP for the RL image classification problem, each episode, an image $x \in \mathcal{X}$ is sampled randomly from the dataset, and a label $y \in \mathcal{Y}$ sampled randomly for this image from the distribution $p(y|x, \mathcal{D})$. This label is *held fixed* for the entire episode. For notation, let $Y = \{1, \ldots, d\}$, so that a label distribution $p(y|x)$ can be written as a vector in the probability simplex on $\mathbb{R}^d$. We emphasize two settings: $\gamma = 0$ (the supervised learning setting), and $\gamma = 1$ (an RL setting), where the expected return of an agent is the average number of incorrect guesses made.

### A.2.1 Memory-based policy

Since the optimal memory-based policy in a POMDP is deterministic [47], we restrict ourselves to analyzing the performance of deterministic memory-based policies. In the following we will narrow the search space even further.

Since the episode ends after the agent correctly classifies an image and the reward structure incentives the agent to solve the task as quickly as possible, an agent acting optimally will never repeat the same action twice. Indeed, the agent will not have the opportunity to repeat the right action twice because the episode would have ended after the first time it tried it. Furthermore, trying a wrong action twice is also not optimal as in incurs addition negative reward. Therefore, we can limit our search space to policies that try each action once. These policies differ by the ordering in which they try each one of these $d$ labels.

At the beginning of every episode, a image $x$ is sampled uniformly at random among all training images and its true label $y$ (during that episode) is sampled from $p(y|x, \mathcal{D})$. Let $\pi$ be policy that tries each of the $d$ actions exactly once in its first $d$ trials. Let $T_y^\pi$ denotes the time when policy $\pi$ tries action $y$. Note that $(T_y^\pi)_{y \in \mathcal{Y}}$ is a permutation. When the label chosen is $y$, the cumulative reward of

$\pi$ for that episode is given by $r = \frac{\gamma^{T_y^{\pi}} - 1}{1 - \gamma}$ and the expected cumulative reward (across episodes) is given by:

$$J(\pi) := \sum_{y \in \mathcal{Y}} p(y|x, \mathcal{D}) \frac{\gamma^{T_y^{\pi}} - 1}{1 - \gamma} = \frac{1}{1 - \gamma} \left( \sum_{y \in \mathcal{Y}} p(y|x, \mathcal{D}) \gamma^{T_y^{\pi}} - 1 \right) \tag{6}$$

From that expression, we see that in order to maximize its expected cumulative reward, a policy $\pi$ has to maximize $\sum_{y \in \mathcal{Y}} p(y|x, \mathcal{D}) \gamma^{T_y^{\pi}}$ which can be interpreted as the dot product of the vector $[p(y|x, \mathcal{D})]_{y \in \mathcal{Y}}$ and $[\gamma^{T_y^{\pi}}]_{y \in \mathcal{Y}}$. By the rearrangement inequality, we know that this dot product is maximized when the components of the vectors are arranged in the same ordering.

If we denote by $y_{(1)}, \dots y_{(d)}$ be the labels sorted in order of probability under the belief distribution: $p(y_{(1)}|x, \mathcal{D}) \geq p(y_{(2)}|x, \mathcal{D}) \geq \cdots \geq p(y_{(d)}|x, \mathcal{D})$. Since $0 < \gamma < 1$ the rearrangement inequality implies that the expected return is maximized when $T_{y_{(t)}}^{\pi} = t$. This corresponds to a policy that tries the labels sequentially from the most likely to the least likely.

### A.2.2 Memoryless policy

In this section, we will derive the optimal memoryless policy. Consider a memoryless policy that takes actions according to the distribution $\pi(\cdot|x)$ for the image $x$. When the true label is $y$ for the image $x$, the number of incorrect guesses is distributed as $\text{Geom}(p = \pi(y|x))$.

When the agent guesses correctly the label $y$ at the $t-$th guess then the cumulative reward is given by $r = -\frac{1-\gamma^t}{1-\gamma}$. This happens with probability $(1 - \pi(y \mid x))^t \times \pi(y \mid x)$. The expected return for policy $\pi$ evaluated on image $x$ is then given by:

$$
\begin{aligned}
J(\pi|x) &= -\sum_{y \in \mathcal{Y}} \sum_{t=0}^{\infty} (1 - \pi(y|x))^t \pi(y|x) \frac{1 - \gamma^t}{1 - \gamma} p(y|x, \mathcal{D}) \\
&= \sum_{y \in \mathcal{Y}} p(y|x, \mathcal{D}) \frac{\pi(y \mid x) - 1}{1 - \gamma(1 - \pi(y|x))}
\end{aligned}
\tag{7}
$$

When $\gamma = 0$ (supervised learning problem), $J(\pi) = \sum_{y \in \mathcal{Y}} p(y|x, \mathcal{D}) (\pi(y \mid x) - 1)$ is a linear function of $\pi$ and as expected, the optimal policy is to deterministically choose the label with the highest probability: $\pi^*(y \mid x) = 1 \left[ y = \arg\max_{y \in \mathcal{Y}} p(y|x, \mathcal{D}) \right]$.

When $\gamma > 0$, the optimal policy is the solution to a constrained optimization problem that can be solved with Lagrange multipliers. When $\gamma = 1$, the optimal policy can be written explicitly as:

$$\pi^*(y \mid x) = \frac{1}{\lambda} \sqrt{p(y|x, \mathcal{D})} \tag{8}$$

where $\lambda$ is a normalization constant.

## B  Theoretical Results

**Proposition 5.1.** *If the true MDP $\mathcal{M}$ is sampled from $\mathcal{P}(\mathcal{M})$, and evidence $\mathcal{D}$ from $\mathcal{M}$ is provided to an algorithm during training, then the expected test-time return of $\pi$ is equal to its performance in the epistemic POMDP $\mathcal{M}^{po}$.*

$$J_{\mathcal{M}^{po}}(\pi) = \mathbb{E}_{\mathcal{M} \sim \mathcal{P}(\mathcal{M})}[J_{\mathcal{M}}(\pi) \mid \mathcal{D}]. \tag{2}$$

*In particular, the optimal policy in $\mathcal{M}^{po}$ is Bayes-optimal for generalization to the unknown MDP $\mathcal{M}$: it receives the highest expected test-time return amongst all possible policies.*

*Proof.* This proposition follows directly from the definition of the epistemic POMDP. If the MDP $\mathcal{M}$ is sampled from $\mathcal{P}(\mathcal{M})$ and $\mathcal{D}$ is witnessed, then the posterior distribution over MDPs is given by $\mathcal{P}(\mathcal{M}|\mathcal{D})$, and the expected test-time return of $\pi$ given the evidence is

$$\mathbb{E}_{\mathcal{M} \sim \mathcal{P}(\mathcal{M})}[J_{\mathcal{M}}(\pi)|\mathcal{D}] := \mathbb{E}_{\mathcal{M} \sim \mathcal{P}(\mathcal{M}|\mathcal{D})}[J_{\mathcal{M}}(\pi)].$$

In the epistemic POMDP, where an episode corresponds to randomly sampling an MDP from $\mathcal{P}(\mathcal{M}|\mathcal{D})$, and a single episode being evaluated in this MDP, the expected return can be expressed identically:

$$J_{\mathcal{M}^{\text{po}}}(\pi) := \mathbb{E}_{\mathcal{M}\sim\mathcal{P}(\mathcal{M}|\mathcal{D})}[\mathbb{E}_{\pi,\mathcal{M}}[\sum_{i=0}^{\infty}\gamma^t r(s_t, a_t)]] = \mathbb{E}_{\mathcal{M}\sim\mathcal{P}(\mathcal{M}|\mathcal{D})}[J_{\mathcal{M}}(\pi)].$$

$\square$

## B.1 Optimal MDP Policies can be Arbitrarily Suboptimal

**Proposition B.1.** *Let $\epsilon > 0$. There exists posterior distributions $\mathcal{P}(\mathcal{M}|\mathcal{D})$ where a deterministic Markov policy $\pi$ is optimal with probability at least $1 - \epsilon$,*

$$P_{\mathcal{M}\sim\mathcal{P}(\mathcal{M}|\mathcal{D})}\left(\pi \in \arg\max_{\pi'} J_{\mathcal{M}}(\pi')\right) \geq 1 - \epsilon, \tag{9}$$

*but is outperformed by a uniformly random policy in the epistemic POMDP: $J_{\mathcal{M}^{po}}(\pi) < J_{\mathcal{M}^{po}}(\pi_{unif})$.*

*Proof.* Consider two deterministic MDPs, $\mathcal{M}_A$, and $\mathcal{M}_B$ that both have two states and two actions: "stay" and "switch". In both MDPs, the reward for the "stay" action is always zero. In $\mathcal{M}_A$ the reward for "switch" is always 1, while in $\mathcal{M}_B$ the reward for "switch" is $-c$ for $c > 0$. The probability of being in $\mathcal{M}_B$ is $\epsilon$ while the probability of being in $\mathcal{M}_A$ is $1 - \epsilon$. Clearly, the policy "always switch" is optimal in $\mathcal{M}_A$ and so is $\epsilon$-optimal under the distribution on MDPs. The expected discounted reward of the "always switch" policy is:

$$J(\pi_{\text{always switch}}) = (1 - \epsilon)\frac{1}{1 - \gamma} - \epsilon\frac{c}{1 - \gamma} = \frac{1}{1 - \gamma}(1 - (c + 1)\epsilon). \tag{10}$$

On the other hand, we can consider a policy which selects actions uniformly at random. In this case, the expected cumulative reward is

$$J(\pi_{\text{random}}) = (1 - \epsilon)\frac{1}{2}\frac{1}{1 - \gamma} - \epsilon\frac{c}{2}\frac{1}{1 - \gamma} = \frac{1}{2}\frac{1}{1 - \gamma}(1 - (c + 1)\epsilon) = \frac{1}{2}J(\pi_{\text{always switch}}). \tag{11}$$

Thus for any $\epsilon$ we can find a $c > \frac{1}{\epsilon} - 1$ such that both policies have negative expected rewards and we prefer the random policy for being half as negative. $\square$

## B.2 Bayes-optimal Policies May Take Suboptimal Actions Everywhere

We formalize the remark that optimal policies for the MDPs in the posterior distribution may be poor guides for determining what the Bayes-optimal behavior is in the epistemic POMDP. The following proposition shows that there are epistemic POMDPs where the support of actions taken by the MDP-optimal policies is disjoint from the actions taken by the Bayes-optimal policy, so no method can "combine" the optimal policies from each MDP in the posterior to create Bayes-optimal behavior.

**Proposition B.2.** *There exist posterior distributions $\mathcal{P}(\mathcal{M}|\mathcal{D})$ where the support of the Bayes-optimal memoryless policy $\pi^{*po}(a|s)$ is disjoint with that of the optimal policies in each MDP in the posterior. Formally, writing $\text{supp}(\pi(a|s)) = \{a \in \mathcal{A} : \pi(a|s) > 0\}$, then $\forall\mathcal{M}$ with $\mathcal{P}(\mathcal{M}|\mathcal{D}) > 0$ and $\forall s$:*

$$\text{supp}(\pi^{*po}(a|s)) \cap \text{supp}(\pi^*_{\mathcal{M}}(a|s)) = \emptyset$$

*Proof.* The proof is a simple modification of the construction in Proposition 5.1. Consider two deterministic MDPs, $\mathcal{M}_A$, and $\mathcal{M}_B$ with equal support under the posterior, where both have two states and three actions: "stay", "switch 1", and "switch 2". In both MDPs, the reward for the "stay" action is always zero. In $\mathcal{M}_A$ the reward for "switch" is always 1, while in $\mathcal{M}_B$ the reward for "switch" is $-2$. The reward structure for "switch 2" is flipped: in $\mathcal{M}_A$, the reward for "switch 2" is $-2$, and in $\mathcal{M}_B$, the reward is 1. Then, the policy "always switch" is optimal in $\mathcal{M}_A$, and the policy "always switch 2" is optimal in $\mathcal{M}_B$. However, any memoryless policy that takes either of these actions receives negative reward in the epistemic POMDP, and is dominated by the Bayes-optimal memoryless policy "always stay", which achieves 0 reward. $\square$

## B.3 MaxEnt RL is Optimal for a Choice of Prior

We describe a special case of the construction of Eysenbach and Levine [52], which shows that maximum-entropy RL in a bandit problem recovers the Bayes-optimal POMDP policy in an epistemic POMDP similar to that described in the RL image classification task.

Consider the family of MDPs $\{\mathcal{M}_k\}_{k \in [n]}$ each with one state and $n$ actions, where taking action $k$ in MDP $\mathcal{M}_k$ yields zero reward and the episode ends, and taking any other action yields reward $-1$ and the episode continues. Effectively, $\mathcal{M}_k$ corresponds to a first-exit problem with "goal action" $k$. Note that this MDP structure is exactly what we have for the RL image classification task for a single image. Also consider the surrogate bandit MDP $\hat{\mathcal{M}}$, also with one state and $n$ actions, but in which taking action $k$ yields reward $r_k$ with immediate episode termination. The following proposition shows that running max-ent RL in $\hat{\mathcal{M}}$ recovers the optimal memoryless policy in a particular epistemic POMDP supported on $\{\mathcal{M}_k\}_{k \in [n]}$.

**Proposition B.3.** *Let $\pi^* = \arg\max_{\pi \in \Pi} J_{\hat{\mathcal{M}}}(\pi) + \mathcal{H}(\pi)$ be the max-ent solution in the surrogate bandit MDP $\hat{\mathcal{M}}$. Define the distribution $\mathcal{P}(\mathcal{M}|\mathcal{D})$ on $\{\mathcal{M}_k\}_{k \in [n]}$ as $\mathcal{P}(\mathcal{M}_k|\mathcal{D}) = \frac{\exp(2r_k)}{\sum_j \exp(2r_j)}$. Then, $\pi$ is the optimal memoryless policy in the epistemic POMDP $\mathcal{M}^{po}$ defined by $\mathcal{P}(\mathcal{M}|\mathcal{D})$.*

*Proof.* See Eysenbach and Levine [52, Lemma 4.1]. The optimal policy $\pi^*$ is given by $\pi^*(a = k) = \frac{\exp(r_k)}{\sum_j \exp(r_j)}$. We know from Appendix A.2.2 that this policy is optimal for epistemic POMDP $\mathcal{M}^{po}$ when $\gamma = 1$. $\qquad\square$

If allowing time-varying reward functions, this construction can be extended beyond "goal-action taking" epistemic POMDPs to the more general "goal-state reaching" setting in an MDP, where the agent seeks to reach a specific goal state, but the identity of the goal state hidden from the agent [52, Lemma 4.2].

## B.4 Failure of MaxEnt RL and Uncertainty-Agnostic Regularizations

We formalize the remark made in the main text that while the Bayes-optimal memoryless policy is stochastic, methods that promote stochasticity in an uncertainty-agnostic manner can fail catastrophically. We begin by explaining the significance of this result: it is well-known that stochastic policies can be arbitrarily sub-optimal in a single MDP, and can be outperformed by deterministic policies. The result we describe is more subtle than this: there are epistemic POMDPs where any attempt at being stochastic in an uncertainty-agnostic manner is sub-optimal, and *also* any attempt at acting completely deterministically is also sub-optimal. Rather, the characteristic of Bayes-optimal behavior is to be stochastic in *some* states (where it has high uncertainty), and not stochastic in others, and a useful stochastic regularization method must modulate the level of stochasticity to calibrate with regions where it has high epistemic uncertainty.

**Proposition B.4.** *Let $\alpha > 0, c > 0$. There exist posterior distributions $\mathcal{P}(\mathcal{M}|\mathcal{D})$, where the Bayes-optimal memoryless policy $\pi^{*po}$ is stochastic. However, every memoryless policy $\pi_s$ that is "everywhere-stochastic", in that $\forall s \in \mathcal{S} : \mathcal{H}(\pi_s(a|s)) > \alpha$, can have performance arbitrarily close to the uniformly random policy:*

$$\frac{J(\pi_s) - J(\pi_{unif})}{J(\pi^{*po}) - J(\pi_{unif})} < c$$

*Proof.* Consider two binary tree MDP with $n$ levels, $\mathcal{M}_1$ and $\mathcal{M}_2$. A binary tree MDP, visualized in Figure 6, has $n$ levels, where level $k$ has $2^k$ states. On any level $k < n$, the agent can take a "left" action or a "right" action, which transitions to the corresponding state in the next level. On the final level, if the state corresponds to the terminal state (in green), then the agent receives a reward of $1$, and the episode exits, and otherwise a reward of $0$, and the agent returns to the top of the binary tree. The two binary tree MDPs $\mathcal{M}_1$ and $\mathcal{M}_2$ are identical except for the final terminal state: in $\mathcal{M}_1$, the terminal state is the left-most state in the final level, and in $\mathcal{M}_2$, the terminal state is the right-most state. Reaching the goal in $\mathcal{M}_1$ corresponds to taking the "left" action repeatedly, and reaching the goal in $\mathcal{M}_2$ corresponds to taking the "right" action repeatedly. We consider the posterior distribution that places equal mass on $\mathcal{M}_1$ and $\mathcal{M}_2$, $\mathcal{P}(\mathcal{M}_1|\mathcal{D}) = \mathcal{P}(\mathcal{M}_2|\mathcal{D}) = \frac{1}{2}$. A policy that reaches the

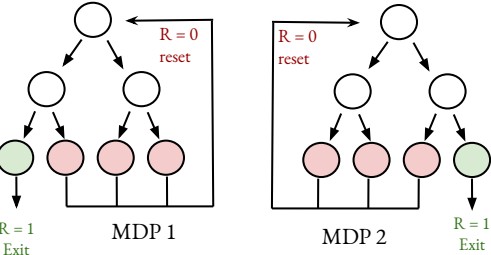

Figure 6: Visual description of Binary Tree MDPs described in proof of Proposition B.4 with depth $n = 3$.

correct terminal state with probability $p$ (otherwise reset) will visit the initial state a $\text{Geom}(p)$ number of times, and writing $\overline{\gamma} := \gamma^n$, will achieve return $\frac{\overline{\gamma}p}{1-\overline{\gamma}+p\overline{\gamma}} = \frac{1}{1+\frac{1}{p}\frac{1-\overline{\gamma}}{\overline{\gamma}}}$.

*Uniform policy:* A uniform policy randomly chooses between "left" and "right" at all states, and will reach all states in the final level equally often, so the probability it reaches the correct goal state is $\frac{1}{2^n}$. Therefore, the expected return is $J(\pi_{\text{unif}}) = \frac{1}{1+2^n\frac{1-\overline{\gamma}}{\overline{\gamma}}}$.

*Bayes-optimal memoryless policy:* The Bayes-optimal memoryless policy $\pi^{*\text{po}}$ chooses randomly between "left" and "right" at the top level; on every subsequent level, if the agent is in the left half of the tree, the agent deterministically picks "left" and on the right half of the tree, the agent deterministically picks "right". Effectively, this policy either visits the left-most state or the right-most state in the final level. The Bayes-optimal memoryless policy returns to the top of the tree a $\text{Geom}(p = \frac{1}{2})$ number of times, and the expected return is given by $J(\pi^{*\text{po}}) = \frac{1}{1+2\frac{1-\overline{\gamma}}{\overline{\gamma}}}$.

*Everywhere-stochastic policy:* Unlike the Bayes-optimal policy, which is deterministic in all levels underneath the first, an everywhere-stochastic policy will sometimes take random actions at these lower levels, and therefore can reach states at the final level that are neither the left-most or right-most states (and therefore always bad). We note that if $\mathcal{H}(\pi(a|s)) > \alpha$, then there is some $\beta > 0$ such that $\max_a \pi(a|s) < 1 - \beta$. For an $\alpha$-everywhere stochastic policy, the probability of taking at least one incorrect action increases as the depth of the binary tree grows, getting to the correct goal at most probability $\frac{1}{2}(1-\beta)^{n-1}$. The maximal expected return is therefore $J(\pi_s) \leq \frac{1}{1+2(\frac{1}{1-\beta})^{n-1}\frac{1-\overline{\gamma}}{\overline{\gamma}}}$

$$J(\pi^{*\text{po}}) = \frac{1}{1+2\frac{1-\overline{\gamma}}{\overline{\gamma}}} \qquad J(\pi_s) = \frac{1}{1+2(\frac{1}{1-\beta})^{n-1}\frac{1-\overline{\gamma}}{\overline{\gamma}}} \qquad J(\pi_{\text{unif}}) = \frac{1}{1+2^n\frac{1-\overline{\gamma}}{\overline{\gamma}}}$$

As $n \to \infty$, $J(\pi^{*\text{po}})$, $J(\pi_s)$ and $J(\pi_{\text{unif}})$ will converge to zero. Using asymptotic analysis we can determine their speed of convergence and find that:

$$J(\pi^{*\text{po}}) \sim \frac{\overline{\gamma}}{2} \qquad J(\pi_s) \sim \frac{\overline{\gamma}}{2(\frac{1}{1-\beta})^{n-1}} \qquad J(\pi_{\text{unif}}) \sim \frac{\overline{\gamma}}{2^n}$$

Using these asymptotics, we find that

$$\frac{J(\pi_s) - J(\pi_{\text{unif}})}{J(\pi^{*\text{po}}) - J(\pi_{\text{unif}})} \sim \frac{1}{(\frac{1}{1-\beta})^{n-1}} = (1-\beta)^{n-1},$$

which shows that this ratio can be made arbitrarily small as we increase $n$. $\square$

*An aside: deterministic policies* While this proposition only discusses the failure mode of stochastic policies, *all* deterministic memoryless policies in this environment also fail. A deterministic policy $\pi_d$ in this environment continually loops through one path in the binary tree repeatedly, and therefore will only ever reach one goal state, unlike the Bayes-optimal policy which visits both possible goal states. The best deterministic policy then either constantly takes the "left" action (which is optimal for $\mathcal{M}_1$), or constantly takes the "right" action (which is optimal for $\mathcal{M}_2$). Any other deterministic policy

reaches a final state that is neither the left-most nor the right-most state, and will always get 0 reward. The expected return of the optimal deterministic policy is $J(\pi_d) = \frac{\overline{\gamma}}{2}$, receiving $\overline{\gamma}$ reward in one of the MDPs, and 0 reward in the other. When the discount factor $\gamma$ is close to 1, the maximal expected return of a deterministic policy is approximately $\frac{1}{2}$, while the expected return of the Bayes-optimal policy is approximately 1, indicating a sub-optimality gap.

$\square$

## B.5 Proof of Theorem 6.1

**Proposition 6.1.** *Let $\pi, \pi_1, \cdots \pi_n$ be memoryless , and define $r_{\max} = \max_{i,s,a} |r_{\mathcal{M}_i}(s,a)|$. The expected return of $\pi$ in $\hat{\mathcal{M}}^{po}$ is bounded below as:*

$$J_{\hat{\mathcal{M}}^{po}}(\pi) \geq \frac{1}{n}\sum_{i=1}^{n} J_{\mathcal{M}_i}(\pi_i) - \frac{\sqrt{2}r_{\max}}{(1-\gamma)^2 n}\sum_{i=1}^{n} \mathbb{E}_{s \sim d_{\mathcal{M}_i}^{\pi_i}}\left[\sqrt{D_{KL}\left(\pi_i(\cdot|s) \,||\, \pi(\cdot|s)\right)}\right], \qquad (3)$$

*Proof.* Before we begin, we recall some basic tools from analysis of MDPs. For a memoryless policy $\pi$, the state-action value function $Q^\pi(s,a)$ is given by $Q^\pi(s,a) = \mathbb{E}_\pi[\sum_{t\geq 0}\gamma^t r(s_t,a_t)|s_0 = s, a_0 = a]$. The advantage function $A^\pi(s,a)$ is defined as $A^\pi(s,a) = Q^\pi(s,a) - \mathbb{E}_{a\sim\pi(\cdot|s)}[Q^\pi(s,a)]$. The performance difference lemma [59] relates the expected return of two policies $\pi$ and $\pi'$ in an MDP $\mathcal{M}$ via their advantage functions as

$$J_{\mathcal{M}}(\pi') = J_{\mathcal{M}}(\pi) + \frac{1}{1-\gamma}\mathbb{E}_{s\sim d_{\mathcal{M}}^{\pi'}}[\mathbb{E}_{a\sim\pi'}[A_{\mathcal{M}}^\pi(s,a)]]. \qquad (12)$$

We now begin the derivation of our lower bound:

$$\begin{aligned} J_{\hat{\mathcal{M}}^{po}}(\pi) &= \frac{1}{n}\sum_{i=1}^{n} J_{\mathcal{M}_i}(\pi) \\ &= \frac{1}{n}\sum_{i=1}^{n} J_{\mathcal{M}_i}(\pi_i) + \frac{1}{n}\sum_{i=1}^{n}[J_{\mathcal{M}_i}(\pi) - J_{\mathcal{M}_i}(\pi_i)] \\ &= \frac{1}{n}\sum_{i=1}^{n} J_{\mathcal{M}_i}(\pi_i) - \frac{1}{n(1-\gamma)}\sum_{i=1}^{n}\mathbb{E}_{s\sim d_{\mathcal{M}_i}^{\pi_i}}\left[\mathbb{E}_{a\sim\pi_i}\left[A_{\mathcal{M}_i}^\pi(s,a)\right]\right] \\ &= \frac{1}{n}\sum_{i=1}^{n} J_{\mathcal{M}_i}(\pi_i) - \frac{1}{n(1-\gamma)}\sum_{i=1}^{n}\mathbb{E}_{s\sim d_{\mathcal{M}_i}^{\pi_i}}\left[\mathbb{E}_{a\sim\pi_i}\left[A_{\mathcal{M}_i}^\pi(s,a)\right] - \mathbb{E}_{a\sim\pi}\left[A_{\mathcal{M}_i}^\pi(s,a)\right]\right] \end{aligned}$$
$$(13)$$

In the last equality we used the fact that $\mathbb{E}_{a\sim\pi}\left[A^\pi(s,a)\right] = 0$. From there we proceed to derive a lower bound:

$$\begin{aligned} J_{\hat{\mathcal{M}}^{po}}(\pi) &= \frac{1}{n}\sum_{i=1}^{n} J_{\mathcal{M}_i}(\pi_i) - \frac{1}{n(1-\gamma)}\sum_{i=1}^{n}\mathbb{E}_{s\sim d_{\mathcal{M}_i}^{\pi_i}}\left[\mathbb{E}_{a\sim\pi_i}\left[A_{\mathcal{M}_i}^\pi(s,a)\right] - \mathbb{E}_{a\sim\pi}\left[A_{\mathcal{M}_i}^\pi(s,a)\right]\right] \\ &\geq \frac{1}{n}\sum_{i=1}^{n} J_{\mathcal{M}_i}(\pi_i) - \frac{2r_{max}}{n(1-\gamma)^2}\sum_{i=1}^{n}\mathbb{E}_{s\sim d_{\mathcal{M}_i}^{\pi_i}}\left[D_{TV}\left(\pi_i(\cdot\mid s);\pi(\cdot\mid s)\right)\right] \\ &\geq \frac{1}{n}\sum_{i=1}^{n} J_{\mathcal{M}_i}(\pi_i) - \frac{\sqrt{2}r_{max}}{(1-\gamma)^2 n}\sum_{i=1}^{n}\mathbb{E}_{s\sim d_{\mathcal{M}_i}^{\pi_i}}\left[\sqrt{D_{KL}\left(\pi_i(\cdot\mid s)\,||\,\pi(\cdot\mid s)\right)}\right] \end{aligned}$$
$$(14)$$

where the first inequality is since $|A_{\mathcal{M}_i}^\pi(s,a)| \leq \frac{r_{\max}}{1-\gamma}$ and the second from Pinsker's inequality. Our intention in this derivation is not to obtain the tightest lower bound possible, but rather to illustrate how bounding the advantage can lead to a simple lower bound on the expected return in the POMDP. The inequality can be made tighter using other bounds on $|A_{\mathcal{M}_i}^\pi(s,a)|$, for example using $A_{\max} = \max_{i,s,a}|A_{\mathcal{M}_i}^\pi(s,a)|$, or potentially a bound on the advantage that varies across state. $\square$

## B.6 Proof of Proposition 6.1

**Proposition 6.2.** *Let $f : \{\pi_i\}_{i \in [n]} \mapsto \pi$ be a function that maps $n$ policies to a single policy satisfying $f(\pi, \cdots, \pi) = \pi$ for every policy $\pi$, and let $\alpha$ be a hyperparameter satisfying $\alpha \geq \frac{\sqrt{2}r_{max}}{(1-\gamma)^2 n}$. Then letting $\pi_1^*, \ldots \pi_n^*$ be the optimal solution to the following optimization problem:*

$$\{\pi_i^*\}_{i \in [n]} = \arg\max_{\pi_1, \cdots, \pi_n} \frac{1}{n} \sum_{i=1}^n J_{\mathcal{M}_i}(\pi_i) - \alpha \sum_{i=1}^n \mathbb{E}_{s \sim d_{\mathcal{M}_i}^{\pi_i}} \left[ \sqrt{D_{KL}\left(\pi_i(\cdot|s) \ || \ f(\{\pi_i\})(\cdot|s)\right)} \right], \quad (4)$$

*the policy $\pi^* \coloneqq f(\{\pi_i^*\}_{i \in [n]})$ is optimal for the empirical epistemic POMDP $\hat{\mathcal{M}}^{po}$.*

*Proof.* By Theorem 6.1 we have that $\forall \alpha \geq \frac{\sqrt{2}r_{max}}{(1-\gamma)^2 n}$:

$$J_{\hat{\mathcal{M}}^{po}}(f(\{\pi_i^*\})) \geq \frac{1}{n} \sum_{i=1}^n J_{\mathcal{M}_i}(\pi_i^*) - \alpha \sum_{i=1}^n \mathbb{E}_{s \sim d_{\mathcal{M}_i}^{\pi_i^*}} \left[ \sqrt{D_{KL}\left(\pi_i^*(\cdot|s) \ || \ f(\{\pi_i^*\})(\cdot|s)\right)} \right]. \quad (15)$$

Now, write $\pi'^* \in \arg\max_\pi J_{\hat{\mathcal{M}}^{po}}(\pi)$ to be an optimal policy in the empirical epistemic POMDP, and consider the collection of policies $\{\pi'^*, \pi'^*, \ldots, \pi'^*\}$. Since $\{\pi_i^*\}$ is the optimal solution to Equation 4, we have

$$\begin{aligned}
J_{\hat{\mathcal{M}}^{po}}(f(\{\pi_i^*\})) &\geq \frac{1}{n} \sum_{i=1}^n J_{\mathcal{M}_i}(\pi'^*) - \alpha \sum_{i=1}^n \mathbb{E}_{s \sim d_{\mathcal{M}_i}^{\pi'^*}} \left[ \sqrt{D_{KL}\left(\pi'^*(\cdot|s) \ || \ f(\{\pi'^*\})(\cdot|s)\right)} \right] \\
&= \frac{1}{n} \sum_{i=1}^n J_{\mathcal{M}_i}(\pi'^*) \\
&= J_{\hat{\mathcal{M}}^{po}}(\pi'^*),
\end{aligned} \quad (16)$$

where the second line here uses the fact that $f(\pi'^*, \ldots, \pi'^*) = \pi'^*$. Therefore $\pi^* \coloneqq f(\{\pi_i^*\})$ is optimal for the empirical epistemic POMDP. $\qquad\square$

# C Procgen Implementation and Experimental Setup

We follow the training and testing scheme defined by Cobbe et al. [16] for the Procgen benchmarks: the agent trains on a fixed set of levels, and is tested on the full distribution of levels. Due to our limited computational budget, we train on the so-called "easy" difficulty mode using the recommended 200 training levels. Nonetheless, many prior work has found a significant generalization gap between test and train performance even in this easy setting, indicating it a useful benchmark for generalization [16, 26, 23]. We implemented LEEP on top of an existing open-source codebase released by Jiang et al. [23]. Full code is provided in the supplementary for reference.

LEEP maintains $n = 4$ policies $\{\pi_i\}_{i \in [n]}$, each parameterized by the ResNet architecture prescribed by Cobbe et al. [16]. In LEEP, each policy is optimized to maximize the entropy-regularized PPO surrogate objective alongside a one-step KL divergence penalty between itself and the linked policy $\max_i \pi_i$; gradients are not taken through the linked policy.

$$\mathbb{E}_{\pi_i}[\min(r_t(\pi)A^\pi(s,a), \text{clip}(r_t(\pi), 1-\epsilon, 1+\epsilon)A^\pi(s,a) + \beta\mathcal{H}(\pi_i(a|s)) - \alpha D_{KL}(\pi_i(a|s) \| \max_j \pi_j(a|s)))]$$

Note that this update in Equation 6 is not exactly solving the optimization problem dictated by Equation 5, since it leverages a one-step estimator for the gradient of the KL penalty in the PG objective, a heuristic known to lead to better optimization in PPO and other deep policy gradient methods. If the proper estimator for the KL penalty is substituted in, then the Bayes-optimal policy in the empirical epistemic POMDP is an optimal solution for Equation 6.

The penalty hyperparameter $\alpha$ was obtained by performing a hyperparameter search on the Maze task for all the comparison methods (including LEEP) amongst $\alpha \in [0.01, 0.1, 1.0, 10.0]$. Since LEEP trains 4 policies using the same environment budget as a single PPO policy, we change the number of

environment steps per PPO iteration from $16384$ to $4096$, so that the PPO baseline and each policy in our method takes the same number of PPO updates. All other PPO hyperparameters are taken directly from [23].

In our implementation, we parallelize training of the policies across GPUs, using one GPU for each policy. We found it infeasible to run more ensemble members due to GPU memory constraints without significant slowdown in wall-clock time. Running LEEP on one Procgen environment for 50 million steps requires approximately 5 hrs in our setup on a machine with four Tesla T4 GPUs.

# D  Procgen Results

## D.1  Main Experimental Results

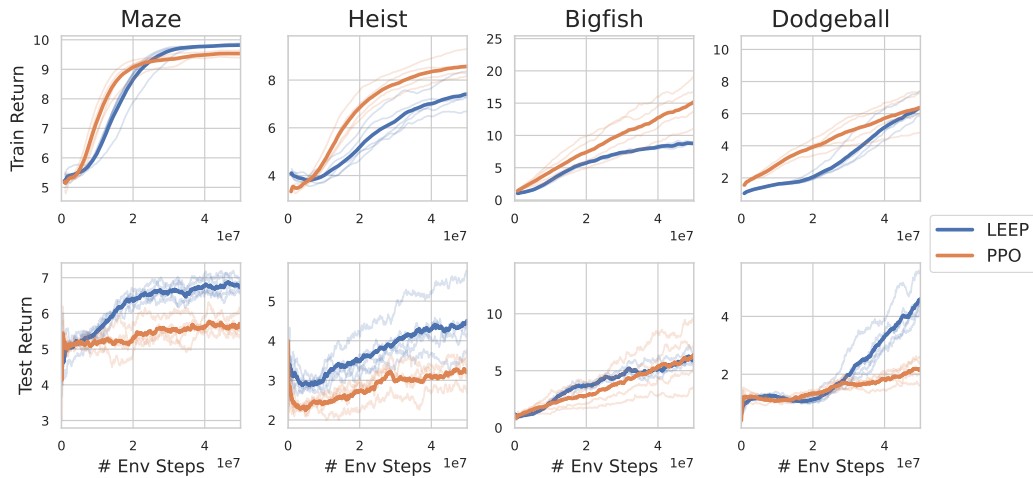

Figure 7: Training (top) and test (bottom) returns for LEEP and PPO on four Procgen environments. Results averaged across 5 random seeds. LEEP achieves equal or higher training return compared to PPO, while having a lower generalization gap between test and training returns.

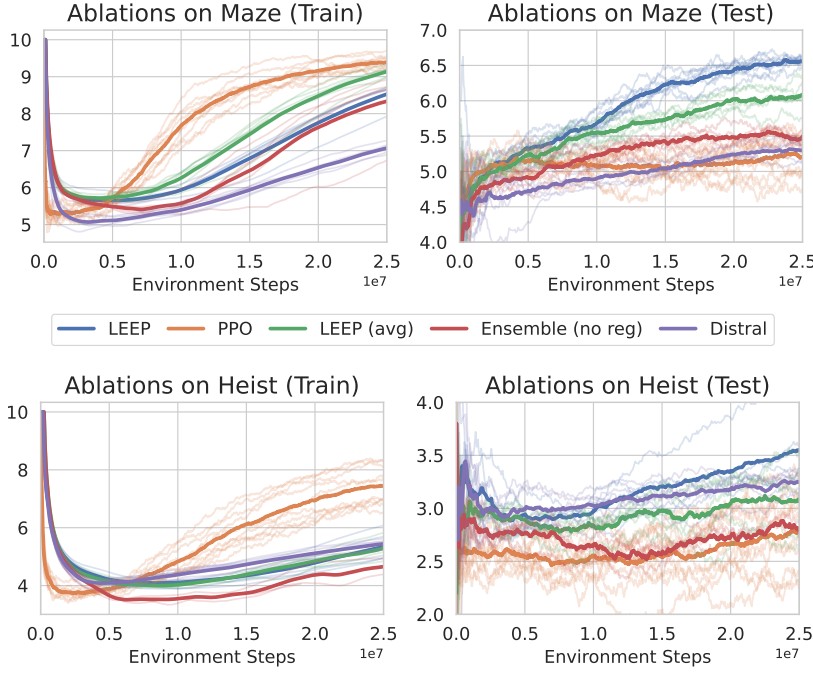

Figure 8: Training and test returns for various ablations and comparisons of LEEP.

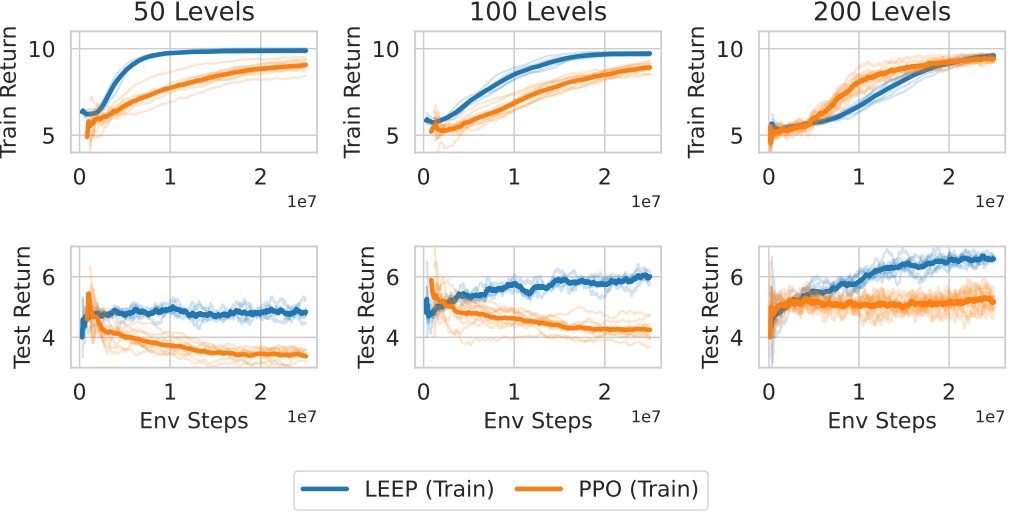

Figure 9: Performance of LEEP and PPO as the number of training levels provided varies. While the learned performance of the PPO policy is worse than a *random policy* with less training levels, LEEP avoids this overfitting and in general, demonstrates a smaller train-test performance gap than PPO.

## D.2 Ablations of LEEP Hyperparameters

**Number of ensemble members (n):** We ran an ablation study on the Procgen Maze task to understand how the number of ensemble members affects the performance of LEEP. We found that for an equal number of gradient steps per ensemble member, LEEP does equally well with $n = 4$ and $8$ ensemble members, but poorly with only 1 or 2 ensemble members (see Figure attached). These results indicate that at least on the Maze task, using n=4 ensemble members is an appropriate balance between approximating the true epistemic POMDP with higher fidelity and minimizing the sample complexity incurred by needing to train more ensemble members with on-policy RL methods.

| # Ensemble members (n) | 1 | 2 | 4 | 8 |
|---|---|---|---|---|
| Maze | $5.11 \pm 0.24$ | $5.85 \pm 0.4$ | $6.53 \pm 0.12$ | $6.91 \pm 0.1$ |

**Penalty coefficient ($\alpha$):** We performed a coarse hyperparameter sweem on the four Procgen domains, testing values $\alpha \in 10^{\{-2,-1,0,1,2\}}$. The results in the table below indicated that performance is roughly coniststent for $\alpha = \{0.1, 1, 10\}$, so while performance does depend on this hyperparameter, it is not overly sensitive, and values around 1 are likely to be a good default initialization.

| Penalty parameter ($\alpha$) | 0 | 0.01 | 0.1 | 1 | 10 | 100 |
|---|---|---|---|---|---|---|
| Maze | 5.78 | 5.725 | $5.94 \pm 0.22$ | $6.53 \pm 0.12$ | $6.54 \pm 0.15$ | 5.7 |
| Heist | 3.3 | 3.4 | $3.2 \pm 0.6$ | $3.73 \pm 0.45$ | $3.65 \pm 0.5$ | 3.15 |
| Bigfish | 1.57 | 2.35 | $2.85 \pm 0.64$ | $4.16 \pm 0.42$ | $3.30 \pm 0.38$ | 1.21 |
| Dodgeball | 0.65 | 0.94 | $0.78 \pm 0.2$ | $1.69 \pm 0.18$ | $1.42 \pm 0.4$ | 1.64 |

## D.3 LEEP and *implicit* partial observability

One common confusion that may arise is that LEEP seeks to overcome partial observability of the contexts, as is done for dynamics generalization in POMDPs (e.g. [60]). This is not the case. Works on dynamics generalization in POMDPs assume that contexts in the true underlying environment are partially observable (e.g. friction coefficients unobserved by a robot without the proper sensors),

and the aim to infer this context using memory. In the epistemic POMDP, *the context is not partially observable*; rather, what is partially observable is how the system dynamics will behave for any provided context, capturing the agent's epistemic uncertainty that stems from the limited training contexts.

We conducted a didactic experiment on Procgen to empirically support the claim that the partial observability modelled by dynamics generalization methods [60] does not replace explicit handling of epistemic uncertainty provided by our method (since this is a different problem). We train a recurrent context encoder that takes in the trajectory seen so far and predicts the identity of the training level. The last hidden layer of this encoder is taken as a "context vector" and fed in as input into a policy alongside the original state, creating an adaptive recurrent policy since this context vector can change through a trajectory. We tested this model on our four Procgen tasks, and made two observations. First, the learned policy, despite being recurrent, does not achieve higher test-time performance than PPO. This is not surprising, because the task is fully observed at training-time. Second, the learned context encoder is able to predict the identity of the training level with > 99% accuracy; that is, the contexts are fully observed and so mechanisms that try to predict the context are unlikely to provide benefit.

The issue is that recurrency and adaptation by themselves are not sufficient to ensure high generalization performance; rather they must be combined with the appropriate model of partial observability that captures the agent's epistemic uncertainty (for LEEP, by statistical bootstrapping on the set of training contexts) to achieve good generalization.

| Test Return after 25M steps | Maze | Heist | Bigfish | Dodgeball |
|---|---|---|---|---|
| PPO | $5.11 \pm 0.24$ | $2.84 \pm 0.46$ | $3.89 \pm 1.64$ | $1.68 \pm 0.33$ |
| PPO with Recurrent Context Encoder | $5.25 \pm 0.5$ | $2.83 \pm 1.04$ | $2.74 \pm 1.1$ | $1.57 \pm 0.3$ |
| LEEP | $6.53 \pm 0.12$ | $3.73 \pm 0.45$ | $4.16 \pm 0.42$ | $1.69 \pm 0.18$ |