# OpenReview forum: "Why Generalization in RL is Difficult: Epistemic POMDPs and Implicit Partial Observability"
_NeurIPS.cc/2021/Conference — NeurIPS 2021 Poster_

### Official Review · Reviewer_C295 · 2021-07-07

**Rating:** 7
**Confidence:** 3

**Summary:**

* This paper uses the insights from Bayesian RL to solve the problem of generalization in RL. In particular, the authors formulated the generalization problem as an epistemic POMDP and proposed an ensemble-based algorithm, called LEEP, to approximately solve the problem.

* The authors empirically demonstrated the performance of LEEP in generalizing to unseen contexts in the Procgen benchmark. They have shown that:
  * LEEP performed better than PPO in Maze, Heist, BigFish, and Dodgeball.
  * An ablation study to show that the better generalization performance of LEEP does not only come from the use of ensembles.
  * An ablation study to show that the better generalization performance of LEEP does come from the inductive bias conferred by the `max_i π_i` link function proposed by the authors.


**Limitations And Societal Impact:**

* I was very confused at the beginning about the difference between this paper and Bayesian RL. Assuming that my understanding of this paper is correct, I think the writing could be improved. Here are my comments:
  * It might be helpful to have a clear paragraph about the `problem definition` of this paper. I have found it confusing between Bayesian RL and the problem that this paper is focusing on.
  * Line 29-45: the `zoo` example conveys the drawback of empirical risk minimization. I understand that the `map` represents empirical risk minimization. However, I am not sure what `peeking through the window` represents? The `anti-empirical risk minimization approach`?
  * Line 29-45: the `zoo` example is good. However, it is an analogy for supervised learning rather than an analogy for RL. Considering that this paper is doing RL, it might be better to use an analogy for RL.
  * Sec. 4 and Fig. 1: the authors try to explain why standard RL methods could fail to generalize. However, the example problem of shoe classification is a contextual bandit problem rather than an MDP problem. Since the authors mentioned `MDP` multiple times in Sec. 4, it might be better to give an example of an MDP rather than a bandit.
  * Sec. 4 and Fig. 1: When I first read Sec. 4 and Fig. 1, I was very confused.
    * Given that methods like UCB can solve bandit problems, when I first read it, I didn't understand why the author said `more sophisticated MDPs that use uncertainty estimates in their construction.` Later on, I understood that the authors are solving a different problem from the problem that UCB is trying to solve.
    * Therefore, I think this section could be clarified a bit to avoid this confusion.
  * Line 228-237: again, I think this is a bandit problem, not an MDP problem.
  * Line 228-237: in this paragraph, the authors use terms like `Bayes-optimal,` which is very confusing to me at first.
    * `Bayes-optimal` refers to optimally solving the trade-off between exploration and exploitation. However, in the problem that this paper is focusing on, there is no exploration.
      * It is very confusing to see `bayes-optimal` and `a = argmax p(y | x, D);` together. Because in Bayesian RL and bandits, `a = argmax p(...)` usually refers to a greedy heuristic to **approximate** the bayes-optimal policy, e.g., UCB. So `a = argmax p(...)` is **not** bayes-optimal.
    * I think maybe the author could define a new concept of optimality for the generalization problem formulated in this paper, rather than using terms in Bayesian RL.
  * Line 228-237: again, the term `POMDP` makes me confusing at first, too.
    * In a POMDP, the belief (or posterior) is updated at every iteration when a new observation comes. However, in the problem formulated in this paper, no exploration (active information gathering) is allowed, and the belief is not updated at all during testing. Thus, I am not sure whether `POMDP` is the correct term. Maybe the authors could use `POMDP` and clarify this point.
  * Line 236: the authors use the word `memoryless` without defining it. I figured it out by reading the appendix. It would be better to define it in advance.
  * Eq. 6: is the policy gradient in Eq. 6 solving the optimal problem? So after convergence, will we get the optimal solution to Eq. 5? It might be better to clarify.


* Minor
  * Line 78: `but also on learning` - `on` is unnecessary.
  * Line 132: `dπ(s) = (1−γ)...` - I don't understand why `(1−γ)` is there (I could be wrong).
  * Line 212: extra space in the beginning.
  * Line 216: extra space in the beginning.
  * Line 233: It might be better to use a larger symbol for the indicator function.

* I didn't check the proofs in Appendix.


# Post-rebuttal
Thank you for your clarification! Now I think the paper is clearer in my mind and I appreciate it more! I have raised my score by 1 point.

**Main Review:**

* This paper proposes a novel problem: the problem of generalization in RL.
  * This problem is closely related to the problem of *Bayesian RL*.
  * However, the difference is that in the authors' problem, the RL agent is **not** allowed to update its belief / posterior after collecting new data. By contrast, a Bayesian RL agent updates its belief / posterior at every iteration.
  * Therefore, Bayesian RL focuses on improving the performance in unseen situations by optimally trading off between exploration vs. exploitation. In contrast, the authors' problem focuses on improving the performance in unseen situations by **exploitation only**.
  * Based on this problem definition, the authors proposed an algorithm to train a policy in a certain way such that this policy can do well in unseen situations even **without any exploration**.


* I think the problem formulation and the algorithm are sound.
  * In line 105, the authors briefly discuss the similarity between domain randomization and the proposed approach. It could make the paper stronger to compare the proposed algorithm with domain randomization in the Procgen benchmark.
  * In line 78, the author says that `Our work highlights the importance of not only (1) finding ways to help neural networks in RL generalize better, but also (2) on learning policies that degrade gracefully when the underlying neural network eventually does fail to generalize.`
    * I am not sure about the difference between (1) and (2). If they are indeed different, has (2) been demonstrated in the results? (it is clear that (1) has been shown in the results).



* I have not seen many other works solving this problem.




**Time Spent Reviewing:**

7

---

> ### Author Response · Authors · 2021-08-10
> **Thank you for your Review!**
>
>
> Thank you for your valuable feedback!
>
> ## 1. Problem Definition
>
> Thank you for your suggestion of delineating the problem statement clearly in a new paragraph. We will update Section 3 with a new paragraph that makes our problem setting more explicit.
>
> ## 2. Section 4: The FashionMNIST Example is an MDP, not a Bandit
>
> To clarify, although the FashionMNIST MDP described in Section 4 appears to be similar to a one-step decision problem (a contextual bandit), it is actually a sequential problem, not a one-step bandit, since the agent makes a second guess for the same image when the first guess is incorrect (and so on), with episodes that can last multiple timesteps. We will add the following text to the description of the FashionMNIST  task  in the first paragraph of Section 4 to make this more clear:
>
> _“This RL problem is near identical to supervised classification, the core distinction being that an agent may interact with the same image over several timesteps in an episode instead of only one attempt as in supervised learning. Note that since episodes may last longer than a single timestep, this problem is not a contextual bandit.”_
>
> ## 3. L29-45: The Zoo Example
>
> _“ I understand that the map represents empirical risk minimization. However, I am not sure what peeking through the window represents?”_
>
> Peeking through the window is an epistemic (information-gathering) action that would never be chosen by standard RL algorithms designed for MDPs (e.g. empirical risk minimization-esque approaches) because it is sub-optimal in all the training environments. In effect, the success of the “peeking” strategy illustrates the need to reason about uncertainty and potentially take information-gathering actions even in the “exploitation” setting for RL. We will also update the example to make it more obvious why this relates to the RL problem instead of the SL setting.
>
> ## 4. Equation 6
>
> _“is the policy gradient in Eq. 6 solving the optimal problem? So after convergence, will we get the optimal solution to Eq. 5? It might be better to clarify.”_
>
> The update in Equation 6 is not exactly solving the optimization problem dictated by Equation 5, since it leverages a one-step estimator for the gradient of the KL penalty in the PG objective, a heuristic known to lead to better optimization in PPO and other deep PG methods (see Schulman et al [1], Sec 3.6 for more discussion). If the proper estimator for the KL penalty is substituted in, then the Bayes-optimal policy in the empirical epistemic POMDP is an optimal solution for Equation 6. We will add discussion in the paragraph L339-342 to clarify this detail.
>
> [1] John Schulman, Xi Chen, Pieter Abbeel. Equivalence Between Policy Gradients and Soft Q-Learning. 2017
>
> ## 5. Clarification about POMDPs
>
> _“again, the term POMDP makes me confusing at first, too.”_
>
> We’d like to clarify the distinction between how the objective is modelled (expected return in a POMDP), and how the objective is solved (learning Markov policies / belief-state policies). When we refer to the POMDP, we are referring to the discounted expected return objective $J_{\mathcal{M}^{po}}(\pi)$, and not the various solution techniques that are often accompanied to solve this objective (e.g. maintaining belief states).
>
> You are correct that it is typical in the POMDP literature to optimize the POMDP expected return objective $J_{\mathcal{M}^{po}}(\pi)$ by learning a belief state and then learning a policy on top of this. However, this is only one possible design choice, and especially in environments where learning belief states may be difficult, a practical alternative is to learn a Markov memoryless policy that does not need to maintain a belief state (for example, [1, 2]). While a Markov memoryless policy may not be optimal for maximizing the POMDP objective  (in the way a belief-state based policy is), even partially maximizing the POMDP objective with a memoryless policy allows the policy to take epistemic uncertainty into account and achieve improved generalization performance.
>
> We will update Section 5 with a paraphrased version of this discussion.
>
> [1] Satinder Singh, Tommi Jaakkola, Michael I Jordan. Learning Without State Estimation in Partially Observable Markovian Decision Processes.
> [2] Guido Montufar, Keyan Ghazi-Zahedi, Nihat Ay. Geometry and Determinism of Optimal Stationary Control in Partially Observable Markov Decision Processes
>
> ## 6. Clarification about Bayes-Optimality
>
> _“It is very confusing to see bayes-optimal and a = argmax p(y | x, D); “_
>
> We will update Section 5 to clarify the definition of Bayes-optimal policy that we use: that a policy $\pi$ is Bayes-optimal for a policy class $\Pi$ if it maximizes the expected return objective in the epistemic POMDP within this class: $\pi \in \arg\max_{\pi \in \Pi} J_{\mathcal{M}^{po}}(\pi)$. If $\Pi$ is not mentioned, it is taken to be the set of all possible policies.
>
> The process-of-elimination policy in L228-237 (the “argmax” policy) being Bayes-optimal is actually an artifact of the very specific structure of the classification MDP in Section 4. As you mention, for a general POMDP, the Bayes-optimal policy is not a simple arg-max, and often requires taking information-gathering actions and updating beliefs. Only for the classification MDP does the Bayes-optimal policy reverts to a simple arg-max operation (argmax amongst all the actions not tried previously within the episode) -- we derive this optimality result in Appendix A.2. We will update the paragraph in L228-237 to emphasize that this is not a general statement about what Bayes-optimal policies look like, and rather only for this specific example.
>
> ## 7. Minor Comments
>
> Thank you, we will update the manuscript to address these comments and typos. We will also define the word memoryless (L236) in the main text to avoid future confusions. On L132, the $(1-\gamma)$ is a normalizing factor that ensures the distribution sums to 1.

---

### Official Review · Reviewer_n4t5 · 2021-07-15

**Rating:** 7
**Confidence:** 4

**Summary:**

The paper addresses the problem of a reinforcement learning generalizing across different dynamics (with the same state/action space). Specifically, the paper frames this problem as a POMDP, where the properties of the environment are encoded as hidden information. An algorithm LEEP is introduced, which learns different policies for different sampled environments, regularized to make the policies similar, and then combines the policy into a single, hopefully robust policy.


**Limitations And Societal Impact:**

The paper does not discuss negative societal impact, but I do not expect this work to have significant societal impact in the foreseeable future.


**Main Review:**

---Update After Author Response---

Much of my original review can be safely ignored, as I had a significant misunderstanding of the premise of the work. It is still included below for the sake of transparency. I have since carefully read the other reviews, the authors' responses to all reviews, and I have carefully re-read the paper.

I now understand that the novel contribution of this paper is the development of a special case of Bayesian RL to the problem of generalizing to unseen states in an MDP. The main insight is that, when the agent has insufficient evidence to extrapolate its policy to unseen states, the agent effectively faces a partially observable problem where the partial observability results from the distribution of possible dynamics at the unknown state, based on the observed evidence. This is indeed, as far as I know, a clever and novel formulation of this issue. For instance, it happens to retroactively justify the common practice of random tie-breaking for action selection. In many typical implementations, if a state is completely unknown, with Q-values all 0, then the agent will act uniformly randomly, which this framework may very well recommend from a more principled position.

Based on my initial confusion, I think I am qualified to suggest that the authors take great care in distinguishing their setting from the better studied setting of generalizing with unobserved context. Even in my re-read, I got tripped up in places and had to take some time to figure out how to reconcile what I was reading with what I understood to be the premise. I think one place where this is particularly important are in Section 5.1, where I think some bigger ballyhoo could be made of the fact that there is only one underlying MDP, that context is observable, and that this distribution is purely a result of the agent's beliefs about the dynamics of that one MDP. Another place where I got tripped up was in Section 4. For a bit I was not sure that the problem is actually fully observable -- I thought that perhaps the fact that the agent has already taken a particular action and received some feedback should be contained in the state description. It took me some time to convince myself that, in the traditional MDP formulation, this problem is simply multiple disconnected single state loops. Perhaps just a diagram of this MDP would make this more immediately clear to readers like myself?

Beyond that, I think the current discussion between the authors and the other reviews covers other suggestions I would make.

---Original Review Below---

---Originality---

This is the area that I am most concerned about with this paper. There are two main contributions here: describing the dynamics generalization problem as a POMDP and the LEEP algorithm. Neither one of them seems particularly novel.

While I can't off the top of my head point to a specific paper that explicitly formulates the generalization problem as a POMDP, it is my impression that it is pretty common to represent the "context" as hidden information. As one example, in "Context-aware Dynamics Model for Generalization in Model-Based Reinforcement Learning" (Lee et al. 2020) they learn a dynamics model that attempts to extract hidden/latent contextual information from recent history and then to use that contextual information to inform decision-making and predictions. They cite prior work that operates on similar ideas and there may be follow-up work as well. They narrow their focus to this particular setting of hidden contextual information, and give that special case its own name (CaDM) rather than casting it in the much broader POMDP framework, but it seems to me that the insight gain from the presentation in this paper is minimal.

The paper seems to suggest that casting this problem as a POMDP offers novel insight into how difficult the dynamics generalization problem is. It even goes so far as to say that there was an "open question" about whether it is difficult to generalize to new dynamics in RL and that this paper provides the answer. I don't find that to be the case. Again, existing work addressing this problem already treats context as hidden state information and already learns policies that are history dependent or stochastic, so noting that this may be necessary seems like old news.

Furthermore, in this paper the formulation as a POMDP doesn't seem to actually go very far. If I am understanding correctly, LEEP is still learning a memoryless policy and does not do any of the Bayesian reasoning about the hidden information that a solution to the POMDP would do. Existing work (e.g. that Lee et al. paper above) seems to already attack this in a more sophisticated way, where the policy can react and change when recent observations offer evidence about what context it might be operating in.

LEEP itself is, as the paper acknowledges, very similar to existing multi-task RL and meta-RL approaches. In a vacuum, I think LEEP does not really add much to what is already out there. The paper suggests that LEEP is notable because it is motivated by this POMDP formulation, but, again, that connection seems tenuous at best.

So I am not sure what this paper substantively adds to the body of work already out there on dynamics generalization in RL. If I am underestimating the novel contribution, then I think the paper needs to be far more explicit about contrasting this formulation/approach with similar, existing ideas.

---Quality---

I don't detect any technical flaws in the theoretical work and the experiments seem well-designed.

As described above, I do think the connection between the POMDP formulation and the algorithm itself is overstated. I don't really see where in this algorithm POMDP-ish inference is supposed to be occurring. For instance, the way the final policy is construction from individual policies is not really ever justified, especially not as a policy that would be suitable for this POMDP.

The experiments do a good job of empirically answering the questions posed. However, a big remaining empirical and conceptual question mark is what are LEEP's relative strengths and weaknesses, compared to existing approaches that aim to improve dynamics generalization in RL?

---Clarity---

I found the paper to be well-written and clearly presented.

---Significance---

Given that I don't see a lot of novel insight in this paper, and given that the algorithm is not adequately contextualized in the related literature on this problem, I would say that the likely impact of this paper would be low.

---Summary of the Review---

The paper is well-written and technically sound, but it seems to me that its main contributions and insights are not sufficiently novel with respect to work already published in this area.


**Time Spent Reviewing:**

3

---

> ### Author Response · Authors · 2021-08-10
> **Thank you for your Review!**
>
> Thank you for your valuable feedback! We believe there is some confusion about the type of partially observability in the epistemic POMDP formulation. While prior work (e.g. Lee et al 2020) has studied learning generalizable policies where the true environment is partially observable (where the context is partially observed), our work focuses on modelling the implicit partial observability that appears in an otherwise fully observable environment due to limited training data effects (e.g., in a contextual MDP, the context is not partially observed, but when generalizing to new contexts, partial observability emerges due to epistemic uncertainty). This is a different problem setting. We discuss this more in #1 below, and include additional experiments with recurrent policies to illustrate this point.
>
> Our response below seeks to clarify
>
> 1. The type of partial observability in the epistemic POMDP and why this is distinct from prior approaches, such as CaDM (Lee et al, 2020) **(w/ new experimental evidence)**, and
> 2. That LEEP can in fact learn recurrent adaptive policies, not just memoryless ones (w/ new experimental evidence), and what the contribution of LEEP is in the context of prior work like Distral.
>
> In the final version of the paper, we will add paraphrased versions of these responses below to better explain what is partially observed in the epistemic POMDP, and more explicitly distinguish our contribution from prior work in generalization for RL. Please let us know if this clarification clears up these issues, or if there are other modifications or experiments you would like us to add.
>
> ## 1. Partial Observability in the Epistemic POMDP and Comparison to CaDM (Lee et al, 2020)
>
> The settings and models of partial observability are entirely different in our paper and ( Lee et al, 2020) and capture different phenomena.
>
> In the (Lee et al, 2020) paper, the true underlying environment is partially observable, since the context is never provided to the agent, and the aim is to infer this context using memory. Even if an infinite number of training contexts were provided to the agent, since the agent does not observe the context in its input, a Markov policy would never perform well even on the training contexts, and would converge to a suboptimal solution. Memory and adaptivity is required in this environment because of this fundamental partial observability that remains no matter how many training environments are observed.
>
> On the other hand, the environments studied in our paper are **fully observed** and the context is observed by the agent. Our focus is on studying generalization to new contexts that are different from those in the training set. In this setting, if the agent is provided with an infinite number of training tasks, a single deterministic Markovian policy can achieve optimal performance (see (Cobbe et al, 2020 Figure 2) for verification on Procgen). However, perhaps surprisingly, when given a finite number of training contexts, such a Markovian policy can be highly suboptimal on test contexts, even if it is good on the training contexts, because the need for generalization from finite training contexts induces implicit partial observability. The partial observability in the resulting “epistemic POMDP” is **not** about the identity of the context, but rather models the agent’s epistemic uncertainty that stems from the limited training contexts. That is, while the main challenge handled by works such as CaDM is to perform accurate context-identification, with the assumption that the optimal policy for _any_ context can be learned via training on the training tasks provided the context identity, the goal in our work is distinct: we show that, because finite training context sets induce partial observability (the epistemic POMDP), it is not possible to learn the optimal behavior for an unseen test context simply by running standard RL methods on the training tasks, even though the context identity is provided as a part of the state. Simply adding memory does not fix this, as we will show below.
>
> As a concrete example of the difference, consider the cartpole task with varying pole lengths from Lee et al, 2020 (Fig 1a). In Lee et al, 2020, the length of the pole is not provided to the RL policy, and it is the role of the recurrent context encoder to infer the length of the pole and act accordingly. The equivalent version of our problem setting for this task would be where the pole length _is provided_ to the agent, but only a select number of pole lengths are provided to the agent during training. What needs to be captured to generalize well in this setting, which is not attained by methods like CaDM, is the agent’s epistemic uncertainty about how the cartpole behaves on pole lengths that have not been seen often during training. By modelling this epistemic uncertainty using the epistemic POMDP, methods like LEEP can generalize well, but methods that seek to predict the context (the pole length) of no value since this length it is trying to infer is already provided to the agent.
>
> **Experiment comparison Recurrent Context Encoder to LEEP:**
>
> We conducted an additional experiment on Procgen to empirically support the claim that the partial observability modelled by dynamics generalization methods like CaDM (Lee et al, 2020) does not replace explicit handling of epistemic uncertainty provided by our method (since this is a different problem). We train a recurrent context encoder that takes in the trajectory seen so far and predicts the identity of the training level. The last hidden layer of this encoder is taken as a “context vector” and fed in as input into a policy alongside the original state, creating an adaptive recurrent policy since this context vector can change through a trajectory. We tested this model on our four Procgen tasks, and made two observations. First, the learned policy, despite being recurrent, does not achieve higher test-time performance than PPO (See table below). This is not surprising, because the task is fully observed at training-time. Second, the learned context encoder is able to predict the identity of the training level with > 99% accuracy; that is, the contexts are fully observed and so mechanisms that try to predict the context are unlikely to provide benefit. The issue is that recurrency and adaptation by themselves are not sufficient to ensure high generalization performance; rather they must be combined with the appropriate model of partial observability that captures the agent’s epistemic uncertainty (for LEEP, by statistical bootstrapping on the set of training contexts) to achieve good generalization.
>
> | Test Return after 25M steps        | Maze         | Heist        | Bigfish      | Dodgeball    |
> |------------------------------------|--------------|--------------|--------------|--------------|
> | PPO                                | 5.11 $\pm$ 0.24 | 2.84 +- 0.46 | 3.89 $\pm$ 1.64 | 1.68 $\pm$ 0.33 |
> | PPO with Recurrent Context Encoder | 5.25 $\pm$ 0.5  | 2.83 $\pm$ 1.04 | 2.74 $\pm$ 1.1  | 1.57 $\pm$ 0.3  |
> | LEEP                               | 6.53 $\pm$ 0.12 | 3.73 $\pm$ 0.45 | 4.16 $\pm$ 0.42 | 1.69 $\pm$ 0.18 |
>
> ## 2. Clarifications about LEEP and significance
>
> **2.1: Memoryless vs. memory-based policies:**
>
> Although in our current presentation, we develop LEEP in the context of memoryless policies, the theoretical results in Section 6 and the LEEP algorithm extend to recurrent policy classes with no modifications, by simply replacing the state $s_t$ with the history $h_t$. We will update the manuscript to make this fact more explicit, and emphasize that our choice of using a memoryless policy over a recurrent one is effectively a “function class” design decision made due to the requirements of the Procgen domain, and not an inherent limiting factor of the LEEP algorithm or framework.
>
> To demonstrate that LEEP can learn adaptive recurrent policies in practice, we ran an experiment on the FashionMNIST task, evaluating 1) an RNN policy trained on the empirical MDP, 2) LEEP with a memoryless Markov policy, and 3) LEEP with a RNN-based policy, presented in the table below. We see that simply training an RNN is insufficient to generalize well, but when the RNN policy class is combined with LEEP (which optimizes the epistemic POMDP), it receives far greater test performance, even higher than w/ the memoryless policy. Qualitatively examining the behavior of the RNN policy trained with LEEP, we see that the policy often change the guessed when the original guess was incorrect, the behavior needed to generalize well.
>
> | Algo        | PPO (Markov) | LEEP (Markov) | PPO (RNN) | LEEP (RNN) |
> |-------------|--------------|---------------|-----------|------------|
> | FashionMNIST Test Return | -2.79        | -1.96         | -2.55     |     -1.45       |
>
> **2.2: Relationship to prior work**
>
> While LEEP follows the well-known algorithmic pattern of training an ensemble of policies regularized to a central policy, what makes LEEP distinct is in what environment each policy $\pi_i$ in the ensemble is trained on. Each ensemble policy $\pi_i$ in LEEP is trained to optimize return in a whole contextual MDP sampled from the posterior distribution $M_i \sim P(\mathcal{M} | \mathcal{D})$, whereas in prior methods like Distral, each policy was trained only for a single context (not a contextual MDP). It is only by combining the ensemble-based regularization in LEEP with appropriate MDPs $M_i$ that captures the agent’s epistemic uncertainty do we learn a Bayes-optimal policy in the epistemic POMDP (Proposition 6.1) that generalizes well in new test environments. We will update the discussion in the final paragraph of Section 6 to make the significance of this contribution more clear.

---

> ### Author Response · Authors · 2021-08-16
> **Follow Up**
>
> We wanted to follow up and ask if our response has helped clarify the significance / originality of the epistemic POMDP and LEEP, and how it differs from prior dynamics generalization work like Lee et al, 2020. We would be grateful if you can let us know whether our responses have addressed your concerns, or if there are remaining concerns that we can clarify.

---

> > ### Comment · Reviewer_n4t5 · 2021-08-17
> > **Re-evaluating**
> >
> > Yes, it's clear to me that I misunderstood the premise of the work. I'm sorry for my mistake and thank you for the clarification!
> >
> > Sorry to leave you hanging -- with a fuller understanding of the premise, I am re-evaluating my assessment. I was waiting until I had formulated a question or new feedback before replying to your comment and/or updating my review.

---

> > > ### Author Response · Authors · 2021-08-24
> > > **Thank you!**
> > >
> > > We really appreciate that you took the time to re-evaluate our submission and provide us with more feedback. We will revise the manuscript to better distinguish our problem setting from the setting of generalization with unobserved context variables, in particular in the introduction, in Section 3, and Section 5.1. Upon your suggestion, we will also add a new diagram (that will be referenced in Section 4 / Figure 1) that shows that the MDP structure of the image classification problem is indeed multiple disconnected single state loops.

---

### Official Review · Reviewer_Vmkh · 2021-07-16

**Rating:** 7
**Confidence:** 4

**Summary:**

The paper investigates the topic of generalization in reinforcement learning. It starts from the observation that even in fully observed environments, a kind of partial observability arises from epistemic uncertainty about the true MDP the agent faces at test time. This observation gives rise to what the paper calls an epistemic POMDP, which has the property that expected return in the epistemic POMDP equals test time agent performance in expectation over the posterior distribution of MDPs (given the prior is accurate).

This insight is used to point out that, being a POMDP, the optimal policy of the epistemic POMDP is in general memory-based, and in the class of memoryless policies, stochastic. In the same vein, even a policy that is optimal on all training contexts can have very poor generalization performance, and the optimal policy for the epistemic POMDP might not be optimal in any single MDP.

The remainder of the paper focuses on memoryless policies in the empirical epistemic POMDP, where the latter arises from a finite number of samples from the MDP posterior. It is shown that one can lower-bound the performance of an ensemble of policies on the empirical epistemic POMDP by their average performance minus a disagreement penalty to the combined policy \pi and that optimizing this bound gives the optimal policy for the empirical epistemic POMDP.

Based on these insights, the paper proposes a practical algorithm (LEEP) that combines bootstrap sampling from the training contexts, choosing max for combining the individual policies, PPO as the base algorithm, and the derived regularization term into an algorithm where the n individual policies are optimized in a round-robin fashion.

The proposed algorithm is evaluated on the procgen benchmark, where it compares favorably to pure PPO and Distral. The algorithm is also shown to avoid overfitting when only few training contexts are available. Finally, an ablation study shows that both the agreement penalty term and the max link function contribute to the performance advantage of LEEP.

**Limitations And Societal Impact:**

Limitations are adequately presented, although it might make sense to be explicit in Proposition 6.1 that "optimal" means "optimal memoryless".

As a very general problem class, RL can be used for both benign and nefarious purposes. Better generalization would make RL better suited for both, but there don't seem to be any points that particularly emphasize the latter.

**Main Review:**

First up, I would like to congratulate the authors on their clear writing. The paper is very easy to follow and especially the first half of the paper is a really enjoyable read. The topic of the paper, generalization, is important to the community and, as far as I know, the challenges faced in generalization settings have not been spelled out this clearly previously.

The paper is split into two parts - a motivational overview part up to and including page 6, while the remaining three pages propose a concrete algorithm based on theoretical insights and evaluate it in experiments.

For me the first half leaves little to be desired. Examples are clearly motivated and illustrative. Using a POMDP perspective to study generalization is an interesting perspective. My only nit-pick would be that the difference to Bayesian RL is over-emphasized. In particular in line L217 the paper seems to imply that Bayesian RL methods are not appropriate because they typically evaluate performance over multiple episodes. However, showing that a method performs well over multiple episodes does not imply that it doesn't on a single one.

The second part suffers as a consequence of the limited space and moves at a fast pace. One large concern is that it seems to me that Proposition 6.1 is over-interpreted. My understanding of (4) is that for any two policies $| J(\pi) - J(\pi_i) | \leq \alpha \mathbb{E}[\mathrm{KL}(\pi, \pi_i)]$. That is, any improvement that a policy $\pi_i$ could have over $\pi$ is smaller than the scaled, average KL. While this is a good motivation for trust-regions, in Proposition 6.1 this is exploited to conclude that by training on (5) we get a Bayes-optimal policy for the epistemic POMDP. However, from my understanding the only reason that we retrieve the optimal policy for the epistemic POMDP is that any improvement on $J_{M_i}(\pi_i)$ by having $\pi_i$ deviate from $f({\pi_i})$ is necessarily smaller than the penalty term. As such, Proposition 6.1 does not make a statement about generalization, but a statement about the strength of the constraint and is largely equivalent to solving $\mathrm{argmax}_{\pi_i} sum J_{M_i}(\pi_i)$ subject to the constraint $\pi_i = f({\pi_i})$. It would be great if the authors could comment on this as part of their rebuttal, since it is a key part of the motivation for LEEP.

A critical component of LEEP seems to be the bootstrap sampling of the contexts. However, the discussion is limited to a single paragraph and there is no ablation study to determine its impact. It would be valueable to elaborate on the importance of this sampling and how the choice $n=4$ for the number of policies impacts experiments.

In general, the experiments focus mostly on a comparison with PPO. While this is in line with the main story about generalization, it would be really interesting how other methods cope with this setting to get an intuition of what the performance limits are here. E.g., including domain randomization (with access to more samples than the finite count) and stateful policies would be very valueable to determine the relative performance of LEEP and how much is (not) lost by focusing exclusively on static policies. After motivating stateful or stochastic policies, it would be great to see the tradeoff analyzed in more detail as part of the experiments. I also find it curious that a method like Distral performs worse, given its similarities to LEEP. Could you hypothesize about why this happens / what the crucial difference is?


### Summary

In summary, I really like the presentation and perspective in the first half, but unfortunately this comes at a cost to the amount of information provided in the second part that proposes an algorithm. I would encourage the authors to trim down the first half (e.g., removing the otter example and focus on the example in Sec. 4) in order to give more room for explanations/interpretations in the second half of the paper. While I do have some concerns about the theoretical motivation and the depth of the experiments that will hopefully be addressed as part of the author feedback, I still think this could make a good paper for NeurIPS.


### Minor & notational comments

- There is some notational confusion between \mathcal{M} and M. In the remainder of the notation sets/spaces are caligraphic, while instances are not, while for the MDP it is flipped. Also in 5.1 \mathcal{M} is defined as a single MDP, while in 6.2 \mathcal{M}_i is a distribution over MDPs. Being clear about what is meant where and being notationally consistent would make the paper more accessible.

**Time Spent Reviewing:**

8

---

> ### Author Response · Authors · 2021-08-10
> **Thank you for your Review!**
>
> Thank you for your valuable feedback! We are glad that you found the paper to be an enjoyable read. We have run experiments investigating the effect of number of ensemble members (n=4) policies (Experiment 1 below) and initial results comparing stateful and static policies (Experiment 2). We discuss these experiments alongside your other specific questions below.
>
> ## 1. Comparison to Bayesian RL
>
> _“My only nit-pick would be that the difference to Bayesian RL is over-emphasized. In particular in line L217…”_
>
> We will update paragraph L216-223 and paragraph L108-124 to mitigate the current over-emphasis on how our problem differs from Bayesian RL. In particular, we will be sure to clarify that, since we study an instantiation of the Bayesian RL formulation, our method can itself be viewed as a particular type of Bayesian RL method, designed specifically for generalization rather than exploration.
>
> ## 2. Proposition 6.1
>
> We have now updated the text to clarify the details regarding Proposition 6.1 and explain how it relates to generalization. When a policy is able to achieve a high return in many different MDPs $M_i$ from the posterior $P(\mathcal{M} | \mathcal{D})$, then it achieves high return in the epistemic POMDP and consequently generalizes well. Proposition 6.1 shows a sufficient condition for when we can expect the policy $f(\{\pi_i\})$ to generalize in such a way.
>
> To sketch in the details, recall that while the optimal way to recover a policy that generalizes well is to optimize performance in the epistemic POMDP, due to practical considerations, we focus instead on learning the optimal policy in the empirical epistemic POMDP, which is supported by a finite number of MDPs, $\\{\mathcal{M}\_i\\}\_{i \in [n]}$. This policy optimization objective can be written as
>
> $\arg\max_{\pi} \sum J_{\mathcal{M}_i}(\pi)$
>
> In our current version, we motivate a trust-region lower bound to $\sum J_{\mathcal{M}_i}(\pi)$ (Theorem 6.1), but as you correctly mention, we can alternatively interpret it as a relaxation of a constrained optimization problem. By reparameterizing $\pi$ as $f(\{\pi_i\})$, this objective can be rewritten as
>
>
> $\arg\max_{\pi_i} \sum J_{\mathcal{M}_i}(\pi_i)$ subject to the constraint $\pi_i = f(\{\pi_i\})$
>
>
> Now, whether interpreting the objective in Proposition 6.1 as a relaxation of a constrained problem or from the lower bound in Theorem 6.1, the proposition explains the same phenomenon: that we can expect the policy $f(\{\pi_i\})$ to generalize when each individual policy $\pi_i$ receives high return in their respective candidate MDPs  from the posterior $\mathcal{M}_i \sim P(\mathcal{M}|\mathcal{D})$ and the policies are close to one another.
>
> ## 3. Importance of # Ensemble Members / Bootstrap Sampling (Experiment 1)
>
> _“It would be valuable to elaborate on the importance of this sampling and how the choice n=4 for the number of policies impacts experiments.”_
>
> During the rebuttal period, we ran an ablation study on the Procgen Maze task to understand how the number of ensemble members affects the performance of LEEP. We found that when controlling for the number of gradient steps taken per ensemble member, LEEP does equally well with n=4 and 8 ensemble members, but poorly with only 1 or 2 ensemble members. There is a tradeoff here, since training more ensemble members using PPO requires collecting more environment samples within the training contexts (higher sample complexity), as visualized in the table below.
>
> | # Ensemble Members (n)   | 1            | 2           | 4            | 8           |
> |--------------------------|--------------|-------------|--------------|-------------|
> | Test Performance on Maze | 5.11 $\pm$ 0.24 | 5.85 $\pm$ 0.4 | 6.53 $\pm$ 0.12 | 6.91 $\pm$ 0.1 |
>
> These results indicate that at least on the Maze task, using n=4 ensemble members is an appropriate balance between fidelity and sample complexity. Understanding alternative approximations to the posterior beyond bootstrapping (for example, maybe instead by learning an ensemble of dynamics models) is an interesting direction, but we believe it to lie outside the scope of the current work. We have added this as an avenue for future work.
>
> ## 4. Stateful vs Static Policies (Experiment 2)
>
> _"stateful policies … how much is (not) lost by focusing exclusively on static policies. After motivating stateful or stochastic policies, it would be great to see the tradeoff..."_
>
> We performed a comparison between stateful and static policies on the FashionMNIST task, running LEEP using both a Markovian and recurrent policy class. Taking epistemic uncertainty into account allows LEEP to improve over PPO for both policy classes, with the recurrent policy trained by LEEP performing marginally better than the Markov one (table below). Qualitatively examining the policies, this performance gap may be explained by the fact that the recurrent policy often changes the guessed label immediately after an incorrect guess, while the stochastic policy must wait for random sampling to choose a different label, which takes longer. Our preliminary experiments on Procgen indicated learning instabilities when training stateful policies on Procgen, so we were unable to perform this comparison on Procgen within the rebuttal period. Nonetheless, we will seek to add a comparison between stateful and static policies on Procgen in the final version of the paper.
>
> | Algo        | PPO (Markov) | LEEP (Markov) | PPO (RNN) | LEEP (RNN) |
> |-------------|--------------|---------------|-----------|------------|
> | FashionMNIST Test Return | -2.79        | -1.96         | -2.55     |     -1.45       |
>
> ## 5. Distral
>
> Although Distral and LEEP look superficially similar, they crucially differ in what environments the policies $\{\pi_i\}$ are trained on. In LEEP, each policy $\pi_i$ is trained on an approximate sample from the posterior distribution over contextual MDPs $\mathcal{M}_i \sim P(\mathcal{M} | \mathcal{D})$, which we estimate via bootstrap resampling. In contrast, the Distral algorithm as originally described by (Teh et al, 2017) trains a separate policy on each individual training context / task. That is, each policy in LEEP is trained on an entire contextual MDP generated by bootstrap resampling (which captures epistemic uncertainty), whereas each policy in Distral is only trained on individual contexts from the set of training contexts (does not necessarily capture epistemic uncertainty). In our experiments, while Distral is able to accelerate learning on the training levels (Appendix D, Figure 7), it does not achieve as high generalization performance as LEEP, likely because the Distral objective does not account for the epistemic POMDP needed to improve generalization.
>
> ## 6. Notation
>
> Thank you for bringing this to our attention; we will update the notation so that calligraphic / boldface styles are consistent, and will add a notation table in the appendix to make the notation easier to understand.

---

### Official Review · Reviewer_pbCR · 2021-07-16

**Rating:** 6
**Confidence:** 3

**Summary:**

The paper considers the problem of generalization in contextual MDPs, where the state of the environment consists of a context that remains fairly constant throughout an episode of the MDP. In such a setting the problem of generalization arises when an RL agent learns on a set of training MDPs (MDPs with contexts sampled from a set of training contexts) and then this trained RL agent must perform well on an unseen context test MDP. This paper underlines the challenges of learning and generalizing in such settings, namely, that methods that are not able to handle epistemic uncertainty fail at generalizing in such settings. The paper formulates this problem of generalization as a special POMDP and proposes a new sampling-based method for approximately solving this POMDP. Experimental results show that the proposed methods beats other baselines on a benchmark suite.

**Limitations And Societal Impact:**

The main limitation of the paper is that it highlights a weakness in the existing method, but then lacks experiments to support that motivation that the proposed method addresses these weaknesses. The paper currently is applied on a benchmark suite of games which I feel does not have a big negative societal impact.

**Main Review:**

Originality: To my knowledge the paper is original

Clarity: The paper is clear and well-written

Significance/Contributions: I think the paper makes significant theoretical, algorithmic and experimental contributions. It highlights the challenges of generalization for an RL agent and formulates the problem as a POMDP, then proposes a new method based on optimizing a proven lower bound on the original objective. The experiments are supportive of the method's ability to generalize on a benchmark suite.

Relevance: Generalization in RL is an important problem to Neurips community and I think the paper is relevant to the Neurips community.

Comments:
- The paper presents a very nice zookeeper example as motivation. Next, the paper also presents a very convincing RL as a classification task as an example. I was looking forward to see the results on the RL as a classification. I dont understand why the paper decided to experiment on a separate benchmark suite. It would have been interesting to see the performance of RL algorithm on the fashion MNIST classification task and to see if the proposed methods is able to come up with a policy that behaves as desired (that is after making an incorrect prediction first the agent changes its prediction)
I think this is a major weakness of the paper since this exactly is the argument used by the paper to show that shortcoming of the current state of the art RL methods. Once this shortcoming is highlighted it is natural for the reader to expect that proposed method will address these shortcomings, but the paper fails to deliver on that.

- Epistemic POMDP: What exactly are the key features of an epistemic POMDP that separates it from the a usual POMDP. Is it s sub-class of a general class of POMDPs or a super-class? A minor point here is that it seems that epistemic POMDP has an underlying policy where the agent predicts the hidden state of the world, specifically for the RL as a classification task. There already exists a class of POMDPs: POMDP with Information reward (and rhoPOMDP) that specifically address this issue. The paper might want to cite and contrast this sub-class of POMDPs.

- I find it a bit surprising that in the experiments a very small value of n = 4 is used. In general, it is normal to use a large value of n for an ensemble method. It would have been nice to see how n matters for this method, even if not on a big deep RL task, but on a small task.

- The paper also introduces a new parameter $alpha$ that balance the trade-off between generalization and overfitting on a context MDP. How to determine this hyperparameter. Experiments studying these hyperparameters are also missing.



**Time Spent Reviewing:**

6

---

> ### Author Response · Authors · 2021-08-10
> **Thank you for your Review!**
>
>
> Thank you for your valuable feedback and a positive assessment of our paper! We have added experiments evaluating our method (LEEP) on the fashionMNIST classification task (Experiment 1 below) and added analyses of the $n$ and $\alpha$ hyperparameters (Experiment 2, 3 below). Below, we address your specific questions and detail the results of these experiments.
>
> ## 1. Evaluation on FashionMNIST (Experiment 1)
>
> We note that this example was meant as a simple didactic thought experiment to illustrate the differences in generalization for RL and supervised learning. However, to address this point, we now have run the LEEP algorithm on the FashionMNIST tasks in two versions: first, where the policy class is Markovian (feedforward network), and second where the policy class is recurrent (LSTM). PPO performs poorly, even with a recurrent adaptive policy class, and LEEP is able to outperform it, achieving highest performance when equipped with a recurrent policy class. Examining the LEEP policies qualitatively, we see that the RNN policy often switches the predicted label when it learns the previous was incorrect, the appropriate behavior needed to generalize well on this task.
>
> | Algo        | PPO (Markov) | LEEP (Markov) | PPO (RNN) | LEEP (RNN) |
> |-------------|--------------|---------------|-----------|------------|
> | FashionMNIST Test Return | -2.79        | -1.96         | -2.55     |     -1.45       |
>
>
> ## 2. Key Features of Epistemic POMDP:
>
> _“What exactly are the key features of an epistemic POMDP that separates it from the usual POMDP. Is it s sub-class of a general class of POMDPs or a super-class?”_
>
> The key feature of an epistemic POMDP that distinguishes it from a usual POMDP is not in its formal structure, but rather that it is an emergent property of studying generalization in _fully observed MDPs_. Unlike other settings where POMDP models are typically employed, partial observability is not something fundamental in the environment, but rather a transient phenomenon that implicitly emerges in the limited training context regime, a distinction illustrated by the following figure: [https://ibb.co/WyxRmz3](https://ibb.co/WyxRmz3)
>
> In terms of formalism, epistemic POMDPs are a sub-class of POMDPs, in which the POMDP state, observation, and transition functions have the particular structure described in L205-211. We will add a paraphrased version of this discussion to the final version of the paper, and add discussion as to how the epistemic POMDP fits into the current taxonomy of POMDPs, including rhoPOMDPs (Araya et al, 2010).
>
> ## 3. Effect of number of ensemble members $n$ (Experiment 2):
>
>  We ran an ablation study on the Procgen Maze task to understand how the number of ensemble members affects the performance of LEEP. We found that for an equal number of gradient steps per ensemble member, LEEP does equally well with n=4 and 8 ensemble members, but poorly with only 1 or 2 ensemble members (see Figure attached). Due to concerns of computational demand, we were unable to investigate with > 8 ensemble members within the rebuttal period, but we will add it to the final version of the paper. These results indicate that at least on the Maze task, using n=4 ensemble members is an appropriate balance between approximating the true epistemic POMDP with higher fidelity and minimizing the sample complexity incurred by needing to train more ensemble members with on-policy RL methods.
>
> | # Ensemble Members (n)   | 1            | 2           | 4            | 8           |
> |--------------------------|--------------|-------------|--------------|-------------|
> | Performance on Maze Task | 5.11 $\pm$ 0.24 | 5.85 $\pm$ 0.4 | 6.53 $\pm$ 0.12 | 6.91 $\pm$ 0.1 |
>
>
> ## 4. Effect of penalty hyperparameter $\alpha$ (Experiment 3):
> During the rebuttal period, we performed a coarse hyperparameter survey on the four Procgen domains, testing values of the penalty hyperparameter $\alpha \in [0, 0.01, 0.1, 1, 10, 100]$ (See Table below). In general, we found that values of $\alpha \in [0.1, 1, 10]$ tended to do well across all the games, indicating that while performance does depend on this hyperparameter, it is not overly sensitive, and that values around 1 are likely to be a good default initialization.
>
> | Penalty parameter ($\alpha$) | 0    | 0.01  | 0.1          | 1             | 10           | 100  |
> |----------------------------|------|-------|--------------|---------------|--------------|------|
> | Maze                       | 5.78 | 5.725 | 5.94 $\pm$ 0.22 | 6.53 $\pm$ 0.12  | 6.54 $\pm$ 0.15 | 5.7  |
> | Heist                      | 3.3  | 3.4   | 3.2 $\pm$ 0.60   | 3.73 $\pm$ 0.45  | 3.65 $\pm$ 0.50  | 3.15 |
> | Bigfish                    | 1.57 | 2.35  | 2.85 $\pm$ 0.64 | 4.16 $\pm$ 0.42  | 3.30 $\pm$ 0.38 | 1.21 |
> | Dodgeball                  | 0.65 | 0.94  | 0.78 $\pm$ 0.20         | 1.69 $\pm$ 0.18  | 1.42 $\pm$ 0.40          | 1.64 |
>
>
> We will update the table with additional seeds which did not finish running in the rebuttal period.

---

### Author Response · Authors · 2021-08-10
**Response to all Reviewers and the AC**

We thank the reviewers for their positive assessment of our work and helpful suggestions for improvement. Please find responses to specific questions directly commented after each review. Based on suggestions from reviewers, we have also added a number of additional experimental results, which are listed below and detailed in the individual responses.

_Ablation study of # ensemble members:_ Upon suggestions from Reviewers pbCR and Vmkh, we have investigated the performance of LEEP on Procgen Mazes, varying the number of ensemble members in [1, 2, 4, 8].

_Ablation study of penalty hyperparameter:_ Upon suggestion from Reviewer pbCR, we have added an ablation study of the penalty hyperparameter across all of our Procgen environments.

_LEEP with RNN on FashionMNIST:_ Upon suggestions from Reviewers pbCR, Vmkh, and n4t5, we have run the LEEP algorithm on the FashionMNIST task from Section 4, using both recurrent and Markov policy classes.

_Comparison to a Recurrent Context Encoder:_  Upon suggestion from Reviewer n4t5, we have compared to a dynamics generalization method that maintains an recurrent encoder which learns a context vector given the recent history, and seeks to use this context vector to inform the policy.

---

### Decision · Program_Chairs · 2021-09-27

**Decision:**

Accept (Poster)

**Comment:**

The paper provides a conceptually interesting perspective on dynamics generalization in RL by drawing a connection to a special subtype of the Bayesian RL formalism. The work's experiments got sufficiently beefed as a result of the discussion with the reviewers, and now the paper is strong both from the conceptual and empirical standpoints. It is important that the authors pay special attention to incorporating the clarifications they gave in response to reviewers' questions into the final paper version. The paper's impact will heavily depend on its clarity and its success in ensuring that readers will see the distinction of the paper's model from general Bayesian RL, something that initially confused several reviewers.